# Analysis and comparison of the bacterial σ⁵⁴ regulon: Evidence of phylogenetic trends in gene regulation

**Maricela Carrera-Reyna[1], Edna Cruz-Flores[2], Enrique Merino**[1]*

**1** Departamento de Microbiología Molecular, Instituto de Biotecnología, Universidad Nacional Autónoma de México, Cuernavaca, Morelos, México, **2** Departamento de Genética del Desarrollo y Fisiología Molecular, Instituto de Biotecnología, Universidad Nacional Autónoma de México, Cuernavaca, Morelos, México

* enrique.merino@ibt.unam.mx

## Abstract

While the role of σ⁵⁴ in regulating genes involved in nitrogen metabolism, flagellar biosynthesis, and stress responses in Pseudomonadota is well established, its involvement in regulating alternative metabolic pathways and cellular processes in other phyla has been less explored. By employing position-specific scoring matrices (PSSMs) to identify promoter sequences regulated by the σ⁵⁴ factor, we successfully predicted genes under its control across 33 taxonomic classes spanning 16 distinct phyla. For the first time, we conducted a comprehensive statistical assessment of σ⁵⁴ regulation across major bacterial phylogenetic groups. Our findings provide an extensive perspective on the regulatory role of σ⁵⁴ beyond nitrogen metabolism and reveal the different trends in which metabolic and biological processes can be regulated by this sigma factor depending on the phylogenetic group. The main findings of our study are available on the aRpoNDB webpage (https://biocomputo.ibt.unam.mx/arpondb/).

## Introduction

Sigma factors are specialized proteins that determine the specificity of promoter recognition and influence the efficiency of RNA polymerase to initiate gene transcription. All bacteria possess a primary sigma factor that directs the transcription of most of their genes during exponential growth, including those involved in essential housekeeping functions. In *E. coli*, this role is fulfilled by σ⁷⁰. In addition to this main sigma factor, many bacterial species encode one or more alternative sigma factors, which regulate the transcription of genes involved in specific physiological responses. These include adaptation to stress conditions, transitions between growth phases, morphological differentiation, or specialized cellular processes, such as sporulation and germination [1]. The number and diversity of alternative sigma factors vary

**Data availability statement:** The data is available on our webpage at https://biocompu-to.ibt.unam.mx/arpondb/ and at the figshare repository at https://figshare.com/projects/Analysis_and_comparison_of_the_bacterial_54_regulon/251936

**Funding:** Funder Name: DGAPA – Universidad Nacional Autónoma de México (UNAM) Grant Number: IN222423 Grant Recipient: Dr. Enrique Merino (E.M.) Additionally, M.C. is a Ph.D. student in the Programa de Doctorado en Ciencias Bioquímicas at UNAM and is supported by a CONACYT fellowship (731966).

**Competing interests:** The authors have declared that no competing interests exist.

widely among various bacterial species and typically reflect their ecological niches and lifestyles. For instance, obligate intracellular organisms residing in stable environments often possess only the essential housekeeping sigma factor. Conversely, free-living bacteria, such as *Streptomyces coelicolor*, which experience dynamic and heterogeneous environmental conditions, may encode dozens of alternative sigma factors to facilitate adaptive responses [2].

From an evolutionary and mechanistic perspective, sigma factors can be categorized into two primary groups. The first group is the most widely distributed and abundant, corresponds to the $\sigma^{70}$ family. Members of this sigma factor enhance the specific interaction between RNA polymerase (RNAP) and promoter DNA, which typically include two conserved elements located approximately −35 and −10 nucleotides upstream from the transcription start site. The second group corresponds to the $\sigma^{54}$ family, which is evolutionarily distinct and currently has no known homologs. $\sigma^{54}$ differs from $\sigma^{70}$ in many instances [3]. It binds to RNAP to form a complex with promoter DNA that is transcriptionally inactive. Transition from this close complex to an open complex, required for transcription initiation, depends on the ATPase activity of a family of enhancer-binding proteins (EBPs) [4,5]. These EBPs bind to upstream enhancer-like sequence in the promoter region in response to specific environmental signals or stress conditions. In addition, the $\sigma^{54}$-recognized promoter differs from that of $\sigma^{70}$ by containing two conserved elements located at positions −24 and −12 relative to the transcription start site instead of the −35 and −10 elements typical of $\sigma^{70}$-dependent promoters [6].

The transcriptional regulatory factor $\sigma^{54}$, also known as RpoN, SigL, NtrA, or σN, is broadly distributed across the bacterial kingdom [7,8]. Although $\sigma^{54}$ is typically classified as an alternative sigma factor, its importance varies across different organisms [9]. In certain microorganisms, $\sigma^{54}$ is essential for survival under challenging conditions and plays a central role as a global regulator within complex transcriptional networks that govern diverse cellular processes [9–12]. Notably, deletion of $\sigma^{54}$ is lethal only in *Myxococcus xanthus* [13]. While $\sigma^{54}$ regulates the expression of essential genes, these genes are often subject to additional regulatory inputs, including other sigma factors and regulatory mechanisms that collectively contribute to their overall gene expression [14].

$\sigma^{54}$ has historically been associated with the regulation of nitrogen metabolism, having been initially identified as a positive regulatory factor required for the expression of glutamine synthetase in enterobacteria [15]. Shortly thereafter, its role was expanded to include the regulation of genes involved in amino acid and sugar transport, the catabolism of compounds such as xylene and toluene, the utilization of alternative carbon sources, extracellular alginate production, flagellar synthesis, bacterial growth, and stress responses. In both pathogenic and phytopathogenic bacteria, $\sigma^{54}$ also contributes to virulence by regulating antibiotic resistance, type IV pili synthesis (T4Ps), motility (twitching and swimming) [7,8]. It also regulates the expression of type III secretion systems (T3SS) [16–19] and type VI secretion systems (T6SS), as well as biofilm formation, quorum sensing, and the transcription of genes involved in toxin production, adhesion, and colonization, all of which enhance pathogenicity

[12,16,17,20–29]. More recently, in organisms such as *Clostridioides difficile* (formerly *Clostridium difficile*), a member of the class Clostridia within the phylum Bacillota, σ54 has been shown to play a crucial role in regulating functions related to metabolism, stress responses, and sporulation, a characteristic feature of these bacteria. Members of this class are known for their ability to form into highly resistant dormant forms called endospores, which enables survival under adverse environmental conditions. σ54 governs the expression of genes involved in the initiation of sporulation, septum formation during sporulation, and spore coat development [26].

Transcription driven by the σ54 promoter involves three main components: a) the RNAP-σ54 holoenzyme, b) the σ54 promoter, and c) EBPs (Fig 1).

a) **The RNAP-σ54 holoenzyme.** When concerning the first player, the RNAP-σ54 holoenzyme, it is crucial to underscore the fact that σ54 encompasses two conserved functional regions, region I and region III, both of which are essential for its proper function. At its N-terminal end lies the activator interaction domain (AID), while the core binding domain (CBD) and the DNA-binding domain (DBD) reside in region III at the C-terminal end. The sequence conservation of these domains is captured in hidden Markov models (HMMs) in the Pfam database [30], denoted as the PF00309, PF04963, and PF04552 profile-HMMs. The AID domain plays an essential role in regulating σ54 activity and functions as a molecular switch. In its inactive state, AID inhibits RNA polymerase activity by impeding the spontaneous isomerization of the open promoter complex. However, upon activation triggered by specific molecules (EBPs), AID switches its role to facilitate transcription initiation [31]. Some Neisseria species possess truncated RpoN proteins lacking one or more of the domains, including the DNA-binding domain (DBD). This loss likely explains the observed 80% reduction in σ54-dependent transcripts [32].

b) **The σ54 promoter.** The promoter element is the second player and consists of two highly conserved sequences that are separated by 5 to 7 nucleotides: 5`-TGGCAC-3` and 5`-TGC-3`. Typically, these conserved motifs are centered at −12 and −24 base pairs relative to the transcription start of the target gene [8].

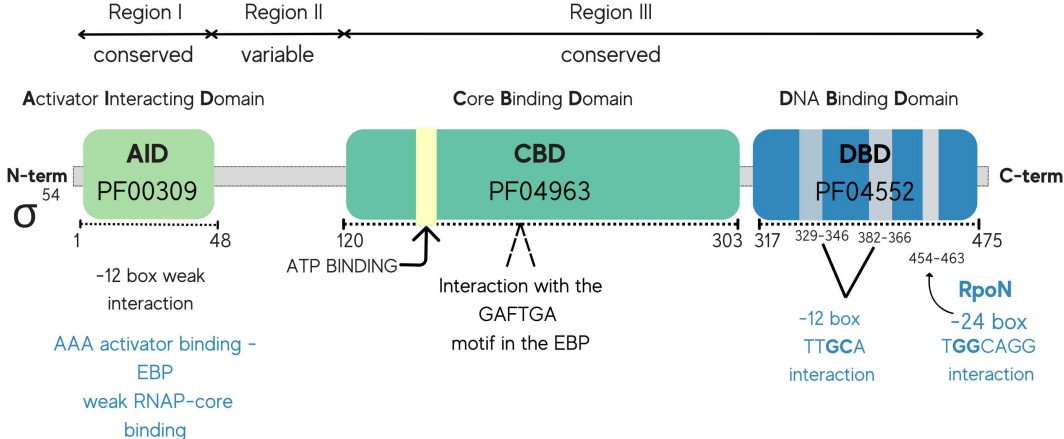

**Fig 1. Schematic representation of the architecture of the functional domains of *Escherichia coli* σ54.** The functional domains of σ54 and their associated roles in transcription initiation are indicated in the figure. Region I is involved in core RNA polymerase, enhancer, and DNA binding and localizes to the activator interacting domain (AID, Pfam number PF00309). Region II, located between Regions I and III, is not essential and is a variable region. Nevertheless, the complete or partial removal of this region can have a significant impact on the activity of σ54, i.e., on DNA binding and holoenzyme formation. Region III is subdivided into several conserved subregions that interact with core RNA polymerase (CBD, Pfam 04963), and DNA at the −12 and the −24 consensus motifs, which are highlighted in the DNA-binding domain (DBD, Pfam PF04552).

**c) EBPs.** The third component in σ⁵⁴-dependent transcription involves EBPs. These proteins have two primary domains: an N-terminal regulatory domain and a C-terminal DNA-binding domain. The N-terminal regulatory domain is responsible for sensing and responding to specific signals or stimuli. EBPs often contain signal perception domains or sensory modules such as CheY-like response regulatory domains, PAS domains (Per-Arnt-Sim), GAF domains (cGMP-specific phosphodiesterases, adenylyl cyclases, and FhlA), PRD modules, and V4R domains. These domains enable EBPs to interact with signaling molecules or undergo conformational changes in response to environmental signals. The C-terminal DNA-binding domain of EBPs recognizes and binds to specific DNA sequences in enhancers. This domain frequently contains a helix-turn-helix (HTH) motif. Although EBPs have different domain structures, maintaining the sequences and structures in their DNA-binding domain is important so that their exact binding sites can be found and contribute to σ⁵⁴-mediated transcriptional activation [33,34].

Each EBP may have unique characteristics, but the "GAFTGA" motif is a defining feature that is invariant in all σ⁵⁴-specific EBPs since this motif is essential for interactions with the σ⁵⁴-RNA polymerase complex and is located within the N-terminal regulatory domain [35,36]. Bacteria most frequently possess one σ⁵⁴ gene, occasionally have two, while multiple EBPs constitute exceptional cases [33]. A single σ⁵⁴ can interact with various EBPs to modulate gene expression in response to specific environmental conditions across different bacterial phyla [17].

Analyses of σ⁵⁴-dependent regulons have been conducted across a range of bacterial groups. These regulons have been effectively characterized using various techniques, including microarrays [37], RNA-seq, and *in silico* approaches. Typically, these studies have been presented independently for each taxonomic group. Some extensively studied groups include Clostridiales [25] and Pseudomonadota, with a focus on classes such as α-Proteobacteria [10,38,39] and γ-Proteobacteria [17,21,28,40–42]. One notable study in this context is the recent comprehensive report by Yu Chao, and their colleagues, representing one of the earliest investigations into the global significance of σ⁵⁴ in a wide range of phytopathogenic bacteria [16]. This in-depth review of σ⁵⁴-dependent regulatory functions, coupled with insights into newly sequenced organisms from less-studied bacterial groups, provides a foundation for further elucidating the intricate regulatory network governed by σ⁵⁴. Nevertheless, a comprehensive global analysis of the regulatory role of σ⁵⁴, including the primary bacterial taxa and newly sequenced organisms from less-studied bacterial groups, has not yet been conducted. Our study, encompassing the examination of σ⁵⁴-dependent regulatory functions in 1,414 organisms from 33 phylogenetic classes spanning 16 distinct phyla, is a significant step toward providing in-depth review of σ⁵⁴-dependent regulatory functions and a more comprehensive understanding of the regulatory role of this sigma factor.

## Methods

### Genome sequence datasets

From a set of 6,889 genomic sequences of bacteria from the KEGG GENOME database [43] we selected nonredundant organisms at the genus phylogenetic level, focusing on those with a greater number of coding genes. Additionally, we included the sequences of the model organisms *Escherichia coli* K-12 MG1655, *Bacillus subtilis* 168, and *Pseudomonas aeruginosa* UCBPP-PA14 as reference points. This selection ensured that a nonredundant and representative dataset was used for our analysis. We clustered these organisms into their respective phylogenetic classes. Initially, 86 phylogenetic classes were considered. Ultimately, we refined our selection to phylogenetic classes containing a minimum of four organisms with an *rpoN* gene. Our final dataset for the study included 1,414 organisms, representing 33 classes from 16 prokaryotic phyla, which are listed in S1 Table.

### Taxonomic classification

The taxonomic assignment of organisms was carried out according to the KEGG GENOME database [43] (https://www.genome.jp/kegg/genome/).

## COG assignments

Proteins from the 1,414 nonredundant organisms in this study were analyzed using a four-step computational procedure to identify their corresponding orthologous groups (Clusters of Orthologous Genes, COGs) [44]. First, we retrieved all proteins for a given COG in the COG database available from the NCBI (https://www.ncbi.nlm.nih.gov/research/cog). Second, we aligned the proteins in each set of COG groups using the MUSCLE 4.1 program [45]. Third, we constructed a hidden Markov model for each COG using the *hmmerbuild* program via the HMMER package v3.3.2 [46]. Fourth, we utilized the HMM matrices created for each COG and the *hmmscan* program, also via the HMMER package v3.3.2., to systematically scan all protein sequences within our set of organisms; the protein domains that exhibited the closest alignment with a COG HMM model was identified. The parameters used for this search included [-E 0.001] to set the E-value threshold in the model output to 0.001; [--domE 0.001] to specify an E-value threshold of 0.001 in the domain output; [--noali] to request that output alignments not be displayed; and [--domtblout] to save a parsable table that includes per-domain information.

## Operon predictions

The transcription units that constitute operons in the set of our studied organisms were predicted based on our previously described algorithm [47], which utilizes an artificial neural network with two input variables: a) intergenic distances between gene pairs transcribed in the same direction and b) the functional relationship between the protein products of adjacent genes as defined in the STRING database [48]. Operon prediction is a key step in our statistical enrichment analysis of genes regulated by $\sigma^{54}$ since it considers all the genes within an operon rather than solely focusing on the first gene located immediately downstream of the $\sigma^{54}$ promoter of the transcription unit. Fig 2 provides an overview of the protocol employed to identify potential $\sigma^{54}$-dependent promoters. The process is structured into three distinct phases.

1. **Data preparation and construction of the seed matrix for $\sigma^{54}$-dependent promoters.** As shown in Fig 2, our protocol for identifying potential $\sigma^{54}$-dependent promoter sequences across phylogenetic groups begins with the construction of a position weight matrix (PWM) to initiate an iterative promoter search. To minimize potential bias toward *E. coli* or other model organisms, we curated a diverse dataset of experimentally validated RpoN-dependent promoter sequences from major regulatory databases, including RegPrecise 3.0 [49], DBTBS [50], and EcoCyc [51], as well as from the comprehensive compilation by Barrios *et al*. (1999) [52]. When promoter sequences appeared in multiple sources, only a single representative was retained to avoid redundancy. The resulting seed dataset comprises 720 experimentally validated $\sigma^{54}$-dependent promoter sequences derived from 56 bacterial species (S2 Table) and served as the basis for constructing an initial positional weight matrix (PWM) that represents the primary consensus sequence motif, also referred to as the "seed matrix", for $\sigma^{54}$-dependent promoters (S3 Table).

2. **Promoter identification.** The second phase involved identifying $\sigma^{54}$-dependent promoters and a cyclic process composed of two steps:

   I. **Pattern-finding.** This step was conducted by employing Motif Alignment and Search Tool (MAST) [53] and a MEME Position-Specific Scoring Matrix (PSSM) [54], matrix for $\sigma^{54}$ promoter sequences was obtained in the pattern discovery step.

   II. **Pattern discovery.** This was performed with a PSSM matrix constructed using Multiple Expectation-Maximization for Motif Elicitation (MEME) software [54], with the $\sigma^{54}$ promoter sequences identified in the MAST search step used as input.

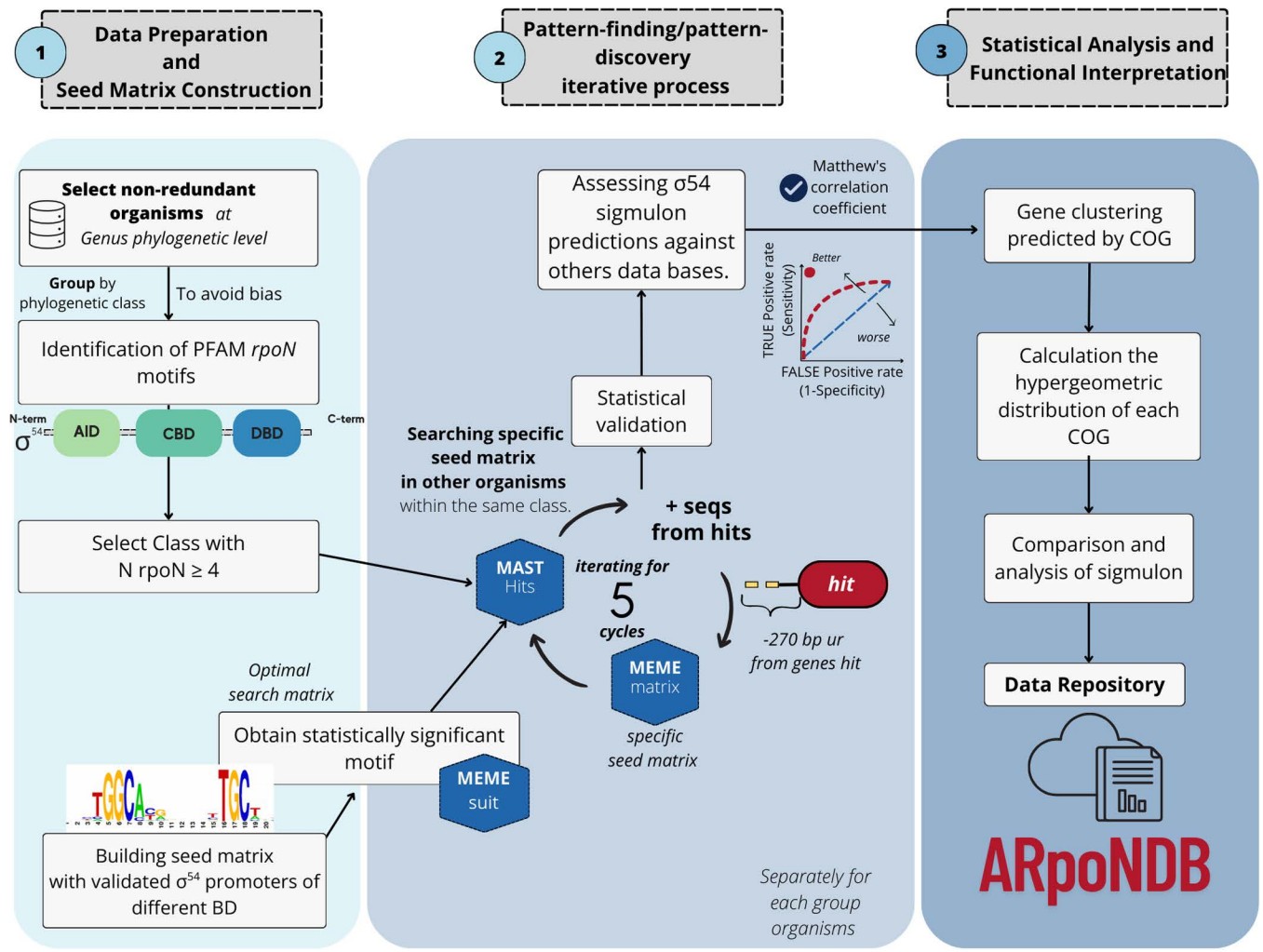

**Fig 2. Methodology overview.** Our methodology was divided into three phases: 1) data preparation and construction of the seed matrix for σ54-dependent promoters, 2) iterative process of pattern finding and pattern discovery, and 3) statistical analysis and functional interpretation. This comprehensive approach ensured the construction of a robust seed matrix that could be used to identify potential σ54-dependent promoters and further investigate their regulatory mechanisms and biological significance.

This cyclic process of pattern-finding/pattern-discovery was independently executed five times to improve the accuracy of the results and was performed independently for each of the 33 phylogenetic classes scrutinized in our study. The first cycle of analysis was conducted with the initial seed matrix.

For MEME analysis, the parameters employed were as follows: [-dna] to specify the usage of the DNA alphabet; [-w 30] to set a 30-nucleotide window size to encompass the two conserved boxes of the bacterial σ54 consensus promoters; [-nmotifs 2] to extract the top two motifs; [-markov_order 1] to indicate the use of a first-order Markov model; [-bfile <bfile>] to specify the background Markov model file name; and [-text] to request the output in text format [54].

Regarding MAST analysis, the parameters utilized were [-norc] to exclude the reverse complement DNA strand scoring and [-bfile <bfile>] to indicate the background Markov model file name. In addition, with the aim of determining the best E-value cutoff values for MAST analysis, the performance of the process was evaluated using ascending E-value cutoffs of 0.625, 1.25, 2.5, and 5. The results from cycles with statistically significant coefficients of sensitivity and specificity

were selected. To assess the accuracy of our prediction program, we compared its results against curated σ54 regulon annotations from two phylogenetically distinct model organisms: *Escherichia coli* (Pseudomonadota) and *Bacillus subtilis* (Bacillota). For this comparison, we used reference data obtained from the EcoCyc [51] and DBTBS [50] databases, respectively. By comparing the real values (genes regulated by σ54, annotated with strong experimental evidence) with the predicted values, we calculated Matthew's Correlation Coefficient (MCC), as a statistical indicator of our protocol´s performance.

3.  **Statistical analysis and functional interpretation.** In the third phase, our focus shifted to performing a comprehensive statistical analysis of the overrepresentation of σ54-regulated genes across the different phylogenetic classes in our study, followed by in-depth interpretation of their biological functions. To achieve this goal, we identified orthologous gene groups in all the genomes based on their orthologous relationships using COG assignments. These assignments were established by employing more than 4,500 Markov models, each corresponding to a COG group and the *hmmscan* program. Next, we employed a PERL program (https://figshare.com/projects/Analysis_and_comparison_of_the_bacterial_54_regulon/251936) that uses the *phyper* and *dhyper* functions from the R package version 4.2.2 to evaluate the hypergeometric distribution, which quantifies the enrichment of each COG protein group within one of the 33 phylogenetic classes under scrutiny. This distribution reveals how effectively σ54 transcribes genes associated with a particular COG across diverse phylogenetic classes compared to the expectations set by the entire gene pool spanning all organisms in the study. This comparative analysis enabled us to identify universally conserved and lineage-specific σ54-dependent genes, offering insights into the regulon's evolutionary conservation.

## Identification of σ54-specific EBPs

To ensure accurate identification of enhancer-binding proteins that interact with σ54 (σ54-specific EBPs), we developed a computational pipeline that integrates both domain-based detection (via PF00158) and conserved motif architecture analysis. The pipeline consists of the following steps:

1.  **Initial EBP Identification:** We searched for the Pfam domain PF00158 (Sigma 54_activat) within the Pfam database [30] and utilized the *hmmsearch* program in the HMMER package v3.3.2 [46], with an E-value threshold of 1e-51 to ensure the inclusion of EBP candidates with high confidence and significant similarity to the PF00158 domain. This domain is conserved in EBP proteins involved in ATP-dependent interactions with σ54.

2.  **Sequence Redundancy Reduction.** To reduce sequence redundancy, the PF00158-containing proteins were clustered using CD-HIT [55] with the following parameters: -c 0.6 (60% identity threshold); -n 3 (word length); -g 1 (accurate clustering mode).

3.  **Motif Discovery.** The resulting non-redundant sequences were analyzed with MEME to identify conserved motifs, using the following settings: -w 12 (motif width of 12); -nmotifs 10 (maximum of 10 motifs); -protein (protein sequence input)

4.  **Motif Scanning Across Proteomes**. We then used MAST to scan all proteomes for the presence of the 10 MEME-defined motifs, using an E-value cutoff of -ev 0.1.

5.  **Filtering for GAFTGA Motif Presence.** Given that the "GAFTGA" motif is experimentally validated as essential for σ54-specific EBP function [35,36], we retained only sequences containing this motif. Notably, these sequences typically also contained all 10 MEME-identified motifs and displayed the most significant E-values in the MAST analysis. Furthermore, the motifs appeared in a consistent and conserved order.

6.  **Final High-Confidence σ54-specific EBP Selection.** We parsed the MAST results to select sequences that contained all 10 motifs arranged in the following conserved order: (2) – (10) – (6) – (3) – (4) – (9) – (7) – (5) – (8) – (1).

Sequences that passed all of the above filters were classified as bona fide σ⁵⁴-specific EBPs.

To evaluate the specificity of our protocol for detecting σ⁵⁴-specific EBP proteins, we used a negative control set consisting of 380 archaeal and 36 fungal (Eukaryota) proteomes, totaling and 1,169,896 protein sequences. No σ⁵⁴-specific EBPs were identified in this group, indicating that our method achieved 100% specificity. The list of organisms used as a negative control is provided in S4 Table.

### Identification of σ⁵⁴ protein domains

The σ⁵⁴ holoenzyme, which is an enhancer-dependent RNA polymerase complex, is characterized by three functional domains:

a) The AID plays a crucial role in facilitating the interaction between σ⁵⁴ and the activator while acting as a transcriptional inhibitor by preventing initiation before activator binding, thereby blocking the entry of the template DNA strand.

b) The DBD directly interacts with the central RNA polymerase.

c) The CBD directly interacts with the core RNA polymerase to form an enhancer-dependent σ⁵⁴ holoenzyme.

To identify these domains in σ⁵⁴ protein sequences, we employed the *hmmsearch* program in the HMMER package v3.3.2 2 [46] with the Pfam motif matrices PF00309, PF04552, and PF04963, using specific E-value threshold values of 1.6e-14, 2.8e-50, and 2.3e-70, respectively. These cutoff values were determined based on the results of the bona fide σ⁵⁴ sequence analysis.

### Identification of σ⁵⁴ protein domains

The 16S rRNA sequences from five randomly selected organisms within each of the 33 phylogenetic classes included in our study were aligned using MAFFT v7.5+ [56], employing the L-INS-i strategy, which is optimized for sequences with globally conserved regions. This multiple sequence alignment was then used to generate a phylogenetic tree with IQ-TREE v2.2+ [57], utilizing its automatic model selection feature to determine the best-fitting evolutionary model. The final tree was visualized using the iTOL web tool [58].

### Web page construction

To visualize the information generated by the methodology proposed in this study, we developed ARpoN-DB, an interactive web-based platform specifically designed to centralize the query and exploration of data related to σ⁵⁴-dependent regulons across multiple bacterial phylogenetic phyla. The interface of ARpoN-DB was implemented using a combination of modern web development technologies including HTML5 for structuring content, including key images and downloadable tables, CSS for responsive design and an intuitive user experience and JavaScript to enable interactive features, such as navigation, dynamic content display, and seamless downloading of data in CSV format. On the backend, PHP processes user requests and manages advanced functionalities. The platform is hosted on an Apache HTTP Server (v2.4), ensuring robust performance, reliability, and efficient resource accessibility. The homepage showcases representative visualizations derived from statistical analyses performed using Perl, Python, and R, providing an overview of the data and its insights. These visualizations and downloadable resources are integral to the platform, facilitating further analysis and exploration by the scientific community.

## Results and discussion

### Phylogenetic distribution of σ⁵⁴

As described in the Methods section, the identification of σ⁵⁴ protein sequences in our studied organisms was based on the detection of Pfam motifs corresponding to its three functional domains: AID (PF00309), DBD (PF04552), and CBD (PF04963). From our dataset of 1,414 organisms of study, we identify 1,150 organisms, which accounted for

approximately 81.3%, contained one or more copies of the *rpoN* gene. These organisms spanned 33 phylogenetic classes across 16 prokaryotic phyla (Fig 3 and S1 Table). In general, most organisms typically possessed a single copy of the *rpoN* gene in their genomes, although significant exceptions to this trend were observed (Fig 3B). Approximately 5.04% of the organisms studied had multiple copies of the σ54 factor gene, as indicated in S5 Table. Notably, six organisms had more than two paralogous copies of σ54. Among these, *Rhodobacter sphaeroides* 2.4.1 (rsp), which belongs to the α-Proteobacteria class, is noteworthy for having up to four *rpoN* genes, as shown in Fig 3B. A previous study conducted by Poggio et al. in 2002 provided evidence of the distinct functions of gene copies within *R. sphaeroides* 2.4.1. Specifically, two of these genes, *rpoN1* and *rpoN2*, were found to play unique roles in gene transcription. RpoN1 was shown to play a crucial role in transcribing a subset of genes under nitrogen-limiting conditions, while RpoN2 was found to be essential for the expression of flagellar genes in this bacterium [59]. These findings offer valuable insights into the functional diversity and specialization of multiple *rpoN* paralogous gene copies within specific bacterial species.

Conversely, Actinomycetota stands out for its near-complete absence of the σ54 factor. Our initial search for *rpoN* genes within this phylum identified only 4 out of 173 representative organisms (<2.31%): *Arthrobacter citreus* NEB 577

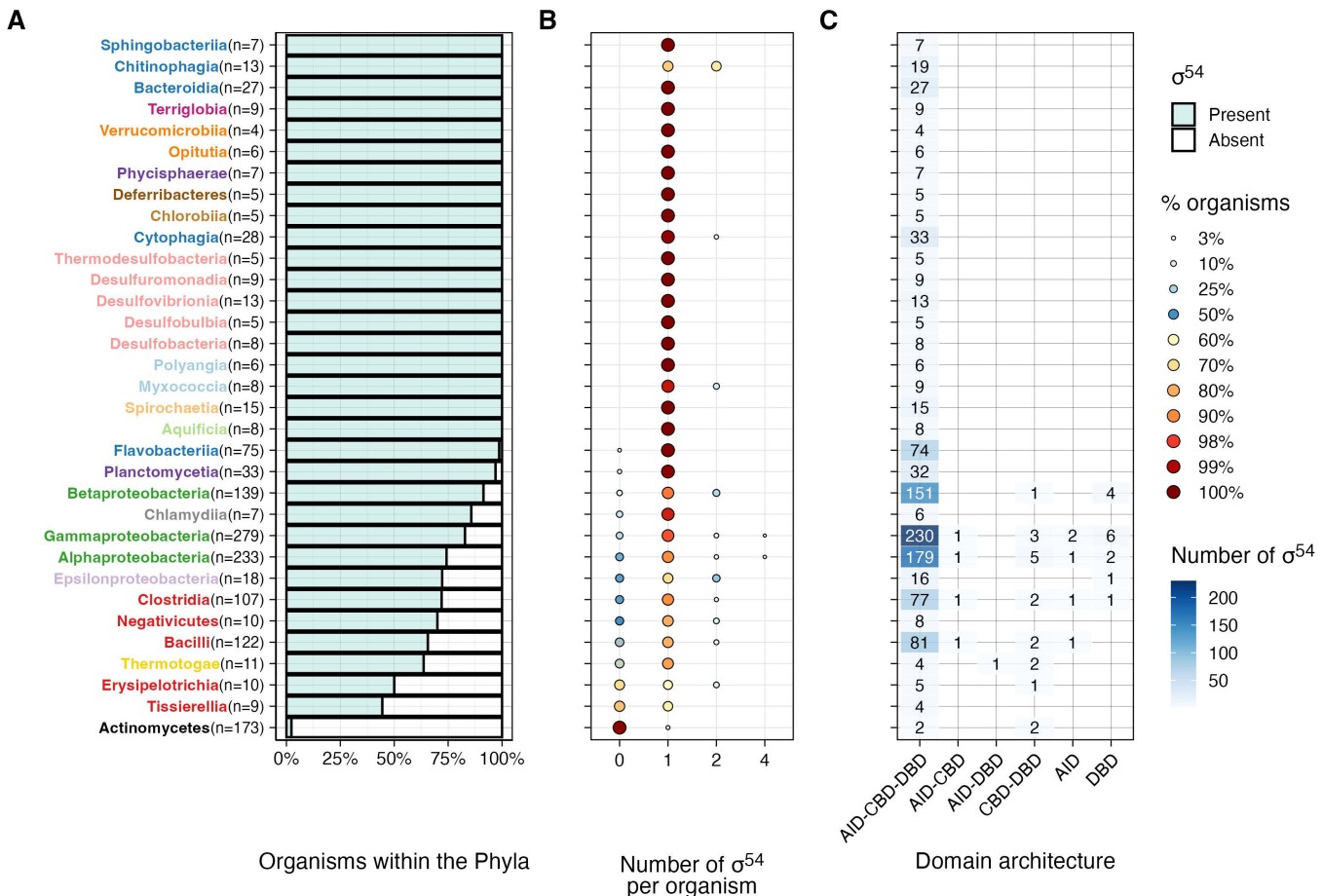

**Fig 3. Analysis of *rpoN* genes in bacteria grouped by phylogenetic class.** A) Percentage of organisms within each phylogenetic class that carry at least one *rpoN* gene encoding the σ54 factor. The number of organisms in the phylogenetic class is denoted by the value "n", shown in parentheses. Phylogenetic classes without an *rpoN* gene are not included. B) Distribution of the number of σ54 genes per organism, with most bacteria having only one copy of the *rpoN* gene. C) Conservation of domains within σ54 proteins, identifying the Pfam domains AID, DBD, and CBD across various bacterial phyla, and illustrating the diverse architectural arrangements of these domains in RpoN proteins.

(acit), *Alloactinosynnema* sp. L-07 (alo), *Modestobacter marinus* BC501 (mmar), and *Citricoccus* sp. SGAir0253−1 (cig) (Fig 3C). However, among these, only *Arthrobacter citreus* NEB 577 encodes an RpoN protein with sequence similarities to other bona fide RpoN proteins and significant Pfam scores across all three conserved domains, PF00309 (AID), PF04552 (CBD), and PF04963 (DBD) (Fig 3C and S6 and S7 Tables). Further analysis, including the construction of a 16S rRNA species phylogenetic tree (S1 Fig) and an extensive review of NCBI database annotations, revealed a significant misclassification: *Arthrobacter citreus* NEB 577, initially assigned to Actinomycetota, aligns more closely with the Bacillota phylum. Taken together, these findings indicate that Actinomycetota effectively lacks a functional RpoN sigma factor. These results align with previous observation regarding the almost inexistent presence of *rpoN* gene in Actinomycetota [9].

A plausible explanation for the absence of the *rpoN* gene in Actinomycetota may be found in their high genomic GC content, which averages around 65%. Additionally, the consensus sequence of σ54-dependent promoters is notably GC-rich, with a GC content of 62.5%. We hypothesize that this overlap in high GC content between the genome and σ54-dependent promoter sequences may impair σ54's ability to distinguish promoters from random genomic sequences reliably. This limitation could impose higher regulatory costs for transcription, contributing to the reduced prevalence of *rpoN* in this phylum. Furthermore, phylogenetic analyses [60], place Actinobacteria within the Terrabacteria branch of the Tree of Life [58], which also includes Cyanobacteria. Interestingly, these phyla rarely possess the *rpoN* gene, with exceptions found in *Gloeocapsa* sp. PCC 7428 and *Leptolyngbya* sp. O-77 (specifically, the Glo7428_4143 and O77CONTIG1_04029 genes, respectively).

The extensive distribution of σ54 factors across various bacterial phyla (as illustrated in Fig 3), combined with the absence of sigma factors in archaea and eukaryotes [2], points to the parsimonious origin of σ54 at the early stage of bacterial evolution, followed by the partial or total loss of this sigma factor in specific bacterial phyla.

## Phylogenetic distribution of σ54-specific EBPs

To identify σ54-specific EBPs within the proteome sequences of our dataset of 1,628 organisms, we used a computational pipeline that integrates both, domain-based detection by searching for the Pfam Sigma54_activat motif (PF00158), and the motif architecture analysis of ten conserved sequence motives present in bona-fide σ54 protein sequences (see Methods section). The distribution of σ54-specific EBPs across different phylogenetic classes and phyla is shown in Fig 4 and detailed in S8 Table. Notably, organisms from the Desulfobacteria class within the Thermodesulfobacteriota phylum possess the most significant number of EBPs genes, including *Desulfosarcina ovata* subsp. sediminis 28bB2T (dov), *Desulfoluna* sp. ASN36 (dek), and *Desulforapulum autotrophicum* HRM2 (dat), with 69, 53 and 52 EBP genes, respectively. In addition, members of the Clostridia class within the Bacillota phylum that displayed a remarkably high abundance of EBPs in their genomes are *Desulfosporosinus orientis* DSM 765 (dor), and *Candidatus Formimonas warabiya* DCMF (fwa) with 67 and 59 paralogous EBPs copies, respectively In contrast, organisms within Actinomycetes, Erysipelotrichia and Chlamydia tended to lack EBPs in their genomes, as shown in Fig 4 and S9 Table. This absence of EBPs corresponds with the notable scarcity of genes encoding σ54 proteins, as illustrated in Fig 3. This analysis highlights the variable distribution of EBPs across different phylogenetic groups, with certain lineages, such as Myxococcota and Desulfobacteria, showing higher counts of these proteins, potentially linked to their distinct genomic and ecological characteristics.

We analyzed the co-occurrence of genes encoding both EBPs and σ54 across all organisms in our study, revealing a consistent correlation: σ54 was always present alongside EBPs (Figs 3 and 4). This finding underscores the indispensable role of EBPs in σ54-dependent transcription. While the necessity of EBPs for isomerization—converting the RNAP-σ54 closed complex into an open complex via ATP hydrolysis—has been previously documented [31,34,61,62], our study provides the first large-scale genomic evidence of this dependency, based on over 1,400 nonredundant genomes.

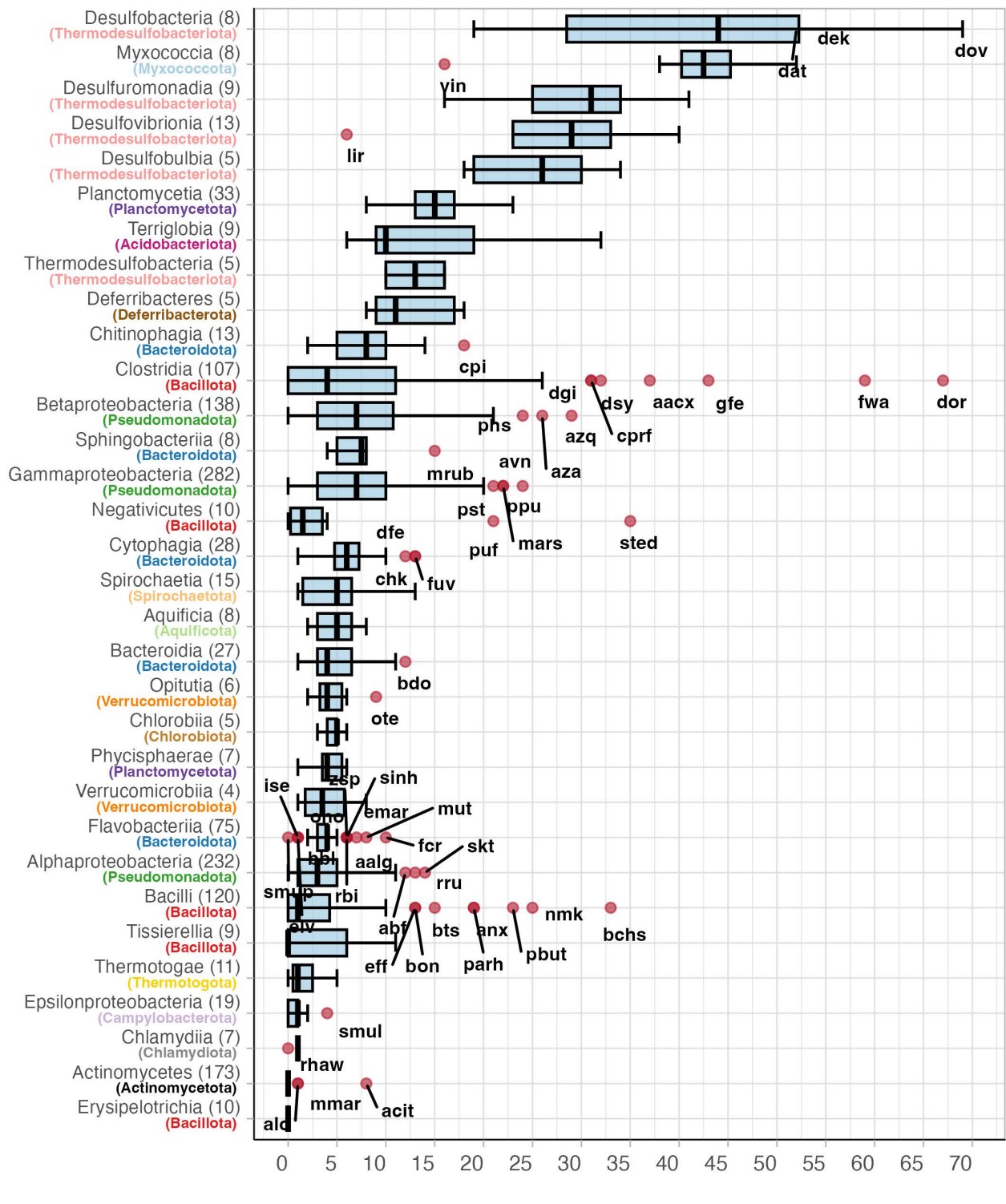

**Fig 4. Distribution of σ⁵⁴-specific EBPs in bacterial classes and phyla.** The distribution of EBPs across bacterial classes and phyla is illustrated using a box and whisker plot. This plot visually represents the number of EBPs per genome within different organisms in each phylogenetic class and phylum. In the plot, a vertical line denotes the median value, while the left and right sides of the box indicate the first and third quartiles, respectively,

thus providing insights into the dispersion of the data. Outliers, represented by red circles outside the whiskers, may indicate unique characteristics or variations in the number of EBP proteins among specific organisms or classes. To facilitate easy comparisons both within and between phyla, each class within the same phylum is assigned the same color. Consistent color coding across phyla ensures uniformity and clarity in visualization. The arrangement of the phyla-class names follows the sequence depicted in Fig 3 for coherence and ease of reference.

### *In silico* identification of σ⁵⁴-dependent promoters

The *in silico* protocol used to identify σ⁵⁴-dependent promoters in our study involved a five-step iterative process of pattern-finding and motif discovery. This process was applied separately to each of the 33 phylogenetic classes to improve precision and minimize bias toward any single model organism. While the accuracy of these predictions may vary, experimental validation is typically required to confirm the functionality of predicted σ⁵⁴ promoters. Nevertheless, our method showed strong performance when tested on two model organisms from distinct phylogenetic groups: *E. coli* (Pseudomonadota) and *B. subtilis* (Bacillota). In *E. coli*, the algorithm achieved a Matthew correlation coefficient (MCC) of 0.672, with a precision of 100%, specificity of 100% and a sensitivity of 45%. For *B. subtilis*, similar values were observed, with an MCC of 0.738, 100% precision of 54%, specificity of 100%, and a sensitivity of 99.8%. The low sensitivity of our method can be attributed to two main factors. First, we used stringent cut-off values to optimize specificity and so be sure that all calculated σ⁵⁴-dependent promoter sites are true positives. Second, the method has limited capacity to predict promoters that significantly deviate from the consensus sequence, TGGCACN5–7TTGCT, such as in the case of the *E. coli*'s promoter sequences of *dcuD* (TGGCAGN7CTCTT), *xapBp3* (TGGCAAN6ACGAT), *glpQp* (TGGACTN6GCTGG). These promoters exhibit specific variations that challenge the predictive accuracy of the algorithm. Such deviations underline the challenge of predicting non-canonical promoters, as the algorithm relies heavily on sequence conservation and spacing constraints characteristic of σ⁵⁴-regulated promoters. The complete set of experimentally validated σ⁵⁴ promoters that our method failed to detect, along with their specific deviations from the consensus sequence, is provided in S10 Table.

It is worth noting that, unlike other computational methods for identifying σ⁵⁴-regulated promoters, which rely on a single position weight matrix (PWM) derived from experimentally characterized σ⁵⁴ promoters such as PromScan (https://molbiol-tools.ca/promscan/), our approach uses a cyclic process in which the initial PWM serves only as a starting point. Subsequent PWMs are independently generated for each phylogenetic class, based on the regulatory regions of the organisms within those classes. We believe that this design, substantially reduces the risk of overfitting. In some cases, bioinformatic methods enhance their predictive accuracy by considering the presence of genes encoding enhancer-binding proteins (EBPs) near DNA sequences that match the σ⁵⁴ promoter consensus, as demonstrated in *Pirellula* species strain 1 studied [27] and *Salmonella enterica* serovar Typhimurium LT2 [63]. However, information on the specificity of EBPs for particular σ⁵⁴-dependent promoters is not available for all the organisms included in our study. More broadly, the task of identifying σ⁵⁴-dependent promoters can be framed as a pattern recognition problem that is well-suited to artificial intelligence (AI) and machine learning (ML) techniques, particularly those capable of learning complex sequence features. A recent study evaluated various model architectures, including random forests, support vector machines, and artificial neural networks, among others, to determine the most effective approach for predicting σ⁵⁴ promoter sequences. The resulting tool, implemented as a web server, bases its predictions on a hybrid architecture combining convolutional and recurrent deep neural networks [64,65].We have compiled the complete set of σ⁵⁴-dependent promoter sequences identified in our study, along with their corresponding σ⁵⁴-regulated operons in a downable zip file available from our web server aRpoNDB (https://biocomputo.ibt.unam.mx/arpondb/) and in the *figshare* repository (https://figshare.com/projects/Analysis_and_comparison_of_the_bacterial_54_regulon/251936). For each gene, the table includes its identifier, the COG annotation of its protein product (classified by cellular or biological process), a functional description, and the source of the promoter sequence, whether derived from the literature, public databases, or identified in this study.

## Phylogenetic diversity in σ⁵⁴-mediated regulation of biological processes

To analyze the taxonomic distribution of genes regulated by the RNAP-σ⁵⁴ complex and uncover principles underlying its evolutionary assembly, we conducted a statistical enrichment analysis of function-related genes transcribed by RNAP-σ⁵⁴ across bacterial phylogenetic classes. This involved calculating the fraction of genes transcribed by RNAP-σ⁵⁴ within each class and clustering genes based on orthologous relationships defined in the COG database [44]. Statistical significance was assessed using the hypergeometric distribution. Results are presented by COG functional categories: 1) metabolism, 2) cellular processes and signaling, 3) information storage and processing, and 4) poorly characterized. Brief explanations of these groups highlight their functional and evolutionary relevance.

**Metabolism.** For historical reasons and due to their significant importance, the first category of genes regulated by σ⁵⁴ that we discuss in our work is the metabolism category. Based on the specific functions of these σ⁵⁴-regulated genes, they can be further categorized into eight groups. The relative enrichment of σ⁵⁴ regulation in these groups is presented in Fig 5 and compiled in S2 Fig. This graph displays the taxonomic distribution of RNAP-σ⁵⁴-regulated genes within metabolic subcategories based on the COG hierarchy. Each data point, represented by a circle, varies in size to reflect the negative log-transformed hypergeometric p-value, indicating the enrichment significance. Subcategories are color-coded for clarity, as detailed in the legend.

**Nitrogen metabolism:** *Nitrogen metabolism* has long been recognized as one of the key processes that is regulated by σ⁵⁴, initially leading to the association of this sigma factor with the name σN. Consistent with previous reports [10,37,66–68] our results indicate that the nitrogen metabolism regulon stands out as highly enriched, particularly in α, β,

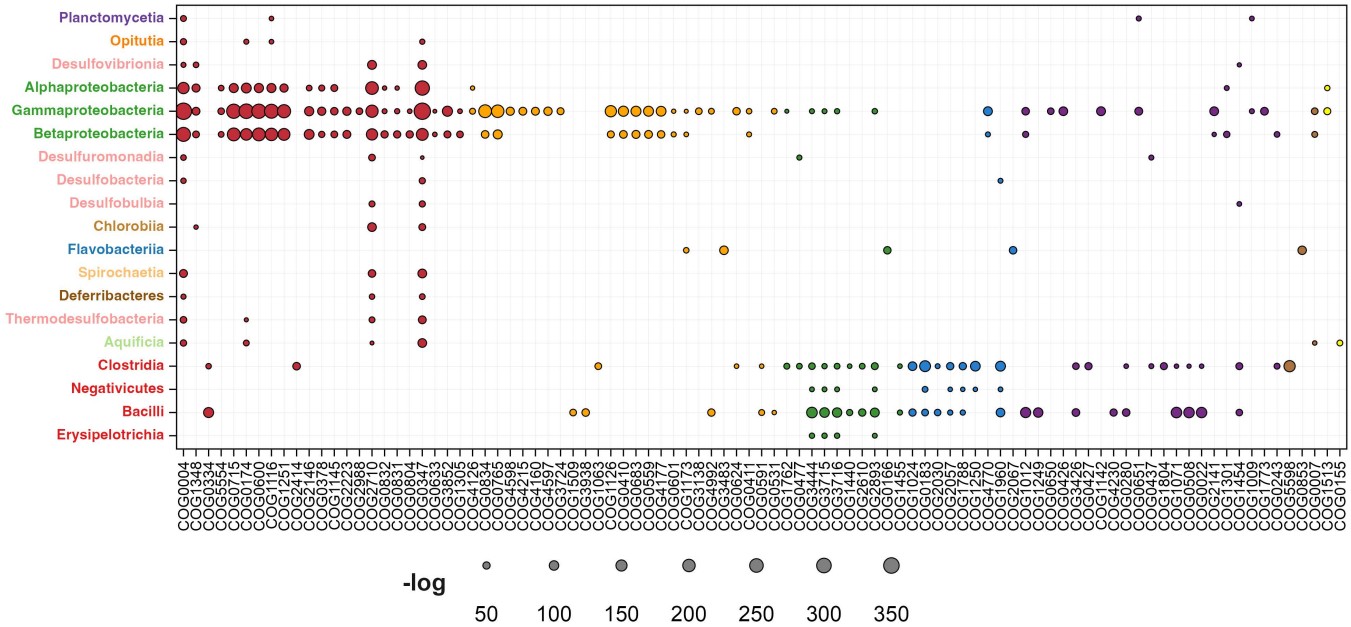

**Fig 5. Functional enrichment analysis of RNAP-σ⁵⁴-regulated genes in the metabolism category.** This graph illustrates the taxonomic distribution of genes regulated by the RNAP-σ⁵⁴ complex across functional subcategories within the metabolic category based on the COG hierarchy. Specifically, this graph showcases the enrichment of predictions within distinct COG subcategories associated with metabolism. Each data point is depicted as a circle, with the size of the circle indicating the negative log-transformed hypergeometric distribution value. This size variation provides a visual representation of the significance of the enrichment for the corresponding COG, thereby facilitating a more straightforward interpretation. To aid in differentiation, colors were assigned to each COG subcategory:. Red: nitrogen metabolism. Orange: amino acid transport and metabolism. Green: carbohydrate transport and metabolism. Blue: lipid transport and metabolism. Purple: energy production and conversion. Brown: coenzyme transport and metabolism. Yellow: inorganic ion transport and metabolism.

and γ-Proteobacteria, its enrichment diminishes to a lesser degree in certain genes related to nitrogen fixation among the Opitutia, Desulfobacteria, Chlorobiia, Spirochaetia, Aquificia, Clostridia, and Bacilli classes. This trend further declined until it was completely absent in the other examined classes, where the RNAP-σ$^{54}$ complex showed a preference for regulating other metabolic pathways and cellular functions.

The first enriched COG group regulated by the RNAP-σ$^{54}$ complex, predominantly in Pseudomonadota, includes components of the nitrogenase enzyme complex critical for nitrogen fixation, specifically the molybdenum-iron nitrogenase protein (COG2710) and the nitrogenase subunit NifH (COG1348).

Glutamine synthetase (COG0174), primarily regulated by RNAP-σ$^{54}$ in Pseudomonadota, catalyzes the synthesis of glutamine from ammonia. Glutamine is crucial for nitrogen metabolism and serves as a key nitrogen carrier and building block for amino acids, nucleotides, and other biomolecules.

A third COG family in Pseudomonadota, regulated by the RNAP-σ$^{54}$ complex, is involved in nitrogen uptake. It includes genes encoding the ammonia permease protein (COG0004) and components of the ABC-type nitrate transport system, such as the permease (COG0600), ATPase (COG1116), and periplasmic (COG0715) components.

The enrichment of transcription regulation by σ$^{54}$ in Pseudomonadota was also observed for the genes encoding NAD(P)H-nitrite reductase (COG1251). This enzyme is essential for the assimilation of inorganic nitrogen by catalyzing the reduction of nitrite, a common form of inorganic nitrogen, to ammonia [69,70].

In addition to genes encoding enzymes and membrane transporters related to nitrogen metabolism, our study primarily identified, within Pseudomonadota, σ$^{54}$-dependent genes encoding regulatory proteins, such as the nitrogen regulatory protein PII (COG0347). Additionally, our findings revealed significant σ$^{54}$ regulation, albeit to a lesser extent, of the urease complex proteins UreA, UreB, UreC, UreD, UreE, UreF, UreF, and UreJ, clustered in the COG0831, COG0832, COG0804, COG0829, COG2370, COG2371, COG0830, and COG0378. The urease enzyme is a key element of nitrogen metabolism since it is responsible for breaking down urea into ammonia, which bacteria utilize as a nitrogen source. This enrichment is predominantly observed in the α and β-Proteobacteria classes.

Conversely, in Bacillota, the COG groups mainly regulated by σ$^{54}$ include glutamate dehydrogenase enzymes involved in ammonia incorporation (COG0334) and arginase and agmatinase enzymes that process nitrogen-rich substrates like arginine and agmatine, aiding in nitrogen recycling and regulation across various metabolic pathways (COG0010).

**Amino acid transport and metabolism:** In the *Metabolism* category, the second group of COGs regulated by the RNAP-σ$^{54}$ complex was associated with *Amino acid transport and metabolism* and is represented by orange circles in Fig 5. Many of these COGs correspond to the ABC-type transport system, a class of membrane transporters found in bacteria that is responsible for the uptake of various nitrogen-containing nutrients, amino acids, ions, and metabolites.

We identified significant enrichment in σ$^{54}$ transcriptional regulation within specific phylogenetic classes. For example, genes encoding ABC-like transport system in several COGs, were found primarily in the α, β, and γ-Proteobacteria classes (COG0410, COG0683, COG0559, COG0765, COG0834, COG1126, COG4177, COG4160, COG4215, COG4597, COG4598, COG0444, COG0601, COG0747, COG1173, COG1176, COG1177, COG3842, COG4608), as well as genes encoding for the hydantoin racemase (COG4126), cysteine sulfinate desulfinase (COG1104), Spermidine/putrescine-binding periplasmic protein (COG0687). Conversely, enrichment of genes predominantly regulated by σ$^{54}$ in the Bacilli class are the N-methylhydantoinase A (COG0145), 4-aminobutyrate aminotransferase (COG0160), proline dehydrogenase (COG0506), proline racemase (COG3938), Lysine 2,3-aminomutase (COG1509), and ornithine/acetylornithine aminotransferase (COG4992).

**Carbohydrate transport and metabolism:** Carbohydrate transport in bacteria involves acquiring and internalizing carbohydrates, with the phosphotransferase system (PTS) being a key mechanism. The PTS allows cells to transport and phosphorylate specific sugars, serving as an essential pathway for obtaining both energy and carbon sources, using energy from phosphoenolpyruvate phosphorylation [19,66,71]. Our study revealed significant enrichment of genes encoding PTS components could be transcribed by the RNAP-σ$^{54}$ complex, which are categorized under carbohydrate transport and

metabolism and represented by green circles in Fig 5. This enrichment was most prominent in classes within the Bacillota phylum, including Bacilli, Clostridia, and Erysipelotrichia. Interestingly, as previously documented, RNAP-σ54 complex also regulates the transcription of PTS component genes in β- [42] and γ-Proteobacteria [63]. These genes include those encoding components for various sugars like mannitol/fructose, mannose/fructose/N-acetylgalactosamine, cellobiose, and galactitol, highlighting the diversity of sugars metabolized through the PTS and σ54's versatility in regulating carbohydrate metabolism.

Beyond the PTS system, other carbohydrate transport and metabolism-related genes regulated by σ54 in the Bacillota phylum included dihydroxyacetone kinase (COG2376) in Clostridia, the H+/gluconate symporter (COG2610), altronate dehydratase (COG2721), and permeases of the major facilitator superfamily (COG0477). Conversely, in β-Proteobacteria, σ54 regulation prominently enriched genes encoding glucose dehydrogenase (COG4993).

**Lipid transport and metabolism:** Lipid metabolism plays a crucial role in biofilm formation, with σ54 acting as a key regulator. It influences motility and metabolic adaptations to nutrient conditions within biofilms, composed of extracellular DNA, lipids, proteins, and polysaccharides [25,72,73]. As depicted in Fig 5 by pink circles, genes associated with lipid transport and metabolism and regulated by σ54 were predominantly enriched in the Clostridia and Bacilli classes within the Bacillota phylum. Among these, several enzymes associated with β-oxidation, such as acyl-CoA dehydrogenases (COG1960) and 3-hydroxyacyl-CoA dehydrogenases (COG1250), were identified as significantly enriched. These enzymes play a central role in breaking down fatty acids into acetyl-CoA for energy production.

Our study identified several lipid metabolism-associated proteins regulated by the RNAP-σ54 complex, including acetyl-CoA acetyltransferase (COG0183), which is involved in both fatty acid synthesis and degradation, enoyl-CoA hydratase/carnitine racemase (COG1024), which supports β-oxidation, and acyl-CoA:acetate/3-ketoacid CoA transferase α and β subunits (COG1788 and COG2057), essential for ketone body utilization.

**Energy production and conversion:** Our analysis of σ54-regulated metabolic genes reveals a diverse array of enzymes critical for energy production and conversion. These processes, depicted by blue circles in Fig 5, are notably enriched in Bacilli and Clostridia. In Bacilli, the pyruvate/2-oxoglutarate dehydrogenase complex (COG1071, COG0508, COG0022, COG1249) connects glycolysis to the Krebs cycle, generating acetyl-CoA, NADH, and FADH2 for ATP synthesis. Additionally, delta 1-pyrroline-5-carboxylate dehydrogenase (COG4230) facilitates proline and arginine catabolism into glutamate, maintaining nitrogen balance. Phosphotransacetylase (COG0280) supports ATP production via substrate-level phosphorylation, emphasizing the regulatory role of σ54 in metabolic processes.

In Clostridia, enzymes such as butyrate kinase (COG3426) and acetyl-CoA hydrolase (COG0427) are crucial for energy metabolism, alongside electron transfer flavoprotein subunits (COG2025 and COG2086) that play a role in electron transport chains. Alcohol dehydrogenase, class IV (COG1454), is found in both Bacilli and Clostridia, where it contributes to the oxidation and reduction of alcohols in energy-yielding pathways.

In Proteobacteria, σ54-dependent regulation affects various classes, including Ni,Fe-hydrogenase small (COG1740) and large (COG0374) subunits, which catalyze hydrogen oxidation or evolution, and Na+/H+-dicarboxylate symporters (COG1301), which import dicarboxylates like succinate and fumarate, important for the TCA cycle [74,75].

Enzymes like NAD-dependent aldehyde dehydrogenases (COG1012) are regulated by σ54 in Bacilli and γ-Proteobacteria, playing a role in oxidizing aldehydes during glycolysis and carbon compound breakdown. Additionally, hydrogenase components (COG4237, COG1142) and formate hydrogenlyase subunits (COG0651) in γ-Proteobacteria emphasize σ54's role in energy metabolism across bacterial clades, highlighting its broad regulatory impact.

**Coenzyme transport and metabolism:** Three COG groups associated with the *Coenzyme transport and metabolism* category are regulated by σ54 and are represented in Fig 5 by purple circles. The first group includes the trimethylamine:corrinoid methyltransferase enzyme, also known as MttB (COG5598), found in the Clostridia class. This enzyme is crucial for coenzyme biosynthesis, specifically in the production of coenzyme B12 (cobalamin), as it facilitates the transfer of a methyl group from trimethylamine (TMA) to a corrinoid protein. The second COG, aspartate 1-decarboxylase (COG0853),

plays a key role in coenzyme A (CoA) biosynthesis by converting L-aspartate into β-alanine, a precursor for pantothenate, an essential component for CoA synthesis. This group of orthologous genes shows the highest enrichment in Flavobacteriia, a class within the phylum Bacteroidota.

**Inorganic ion transport and metabolism:** Fig 5 shows RNAP-σ⁵⁴-mediated regulation of genes in *Inorganic ion transport and metabolism* (yellow circles). Cyanate lyase (COG1513) plays a crucial role in detoxification by breaking down cyanate into carbon dioxide and ammonia, with enrichment in α and γ-Proteobacteria. In Aquificia, the β subunit of sulfite reductase (COG0155) reduces sulfite to sulfide, aiding sulfur incorporation into essential compounds, with significant enrichment highlighting the importance of sulfur metabolism in this group.

**Secondary metabolites biosynthesis/transport and catabolism:** The final category within the metabolism group encompasses the *Biosynthesis, transport, and catabolism* of secondary metabolites. Within this category, several COGs show significant transcriptional regulation by the RNAP-σ⁵⁴ complex. These include COG1335 in α-Proteobacteria, COG0318 and COG1028 in Bacilli, and COG2368 in Clostridia, with the latter being visually represented by the brown circles in Fig 5.

COG1335, encompassing amidases like nicotinamidase, indirectly supports nitrogen metabolism by hydrolyzing nicotinamide into nicotinic acid and releasing ammonia ($NH_3$) as a byproduct. This ammonia contributes to the cellular nitrogen pool, aiding nitrogen assimilation and metabolic processes. While not directly fixing nitrogen, nicotinamidase plays a key role in the $NAD^+$ salvage pathway, particularly in nitrogen-limited environments, where its ability to release ammonia helps maintain nitrogen balance.

The enrichment of acyl-CoA synthetases involved in fatty acid catabolism (COG1318) and dehydrogenases with varying specificities that sustain energy-yielding redox reactions (COG1028) in the Bacilli class underscores their enhanced capabilities for energy production and fatty acid metabolism. These metabolic adaptations are particularly advantageous in nutrient-limited environments or during periods of rapid growth. Together, these COGs form an interconnected network, with acyl-CoA synthetases initiating fatty acid breakdown and dehydrogenases supporting biosynthesis and cellular maintenance, contributing to the adaptability of Bacilli in diverse ecological niches.

COG2368, regulated by σ⁵⁴ primarily in Clostridia, includes flavin-dependent oxidoreductases such as aromatic ring hydroxylase and 4-hydroxybutyryl-CoA dehydratase. These enzymes play roles in aromatic compound degradation, energy production, secondary metabolite synthesis, and detoxification. Notably, 3-hydroxybutyryl-CoA, a fermentation intermediate, is σ⁵⁴-regulated in Clostridial species [25].

**Cellular processes and signaling.** The second functional category examined in our analysis encompasses fundamental cellular processes. The enrichment of genes regulated by σ⁵⁴ is shown in Fig 6 and S3 Fig. This category includes the following classes:

**Cell wall/membrane and envelope biogenesis:** The cell wall, membrane, and envelope constitute a crucial protective barrier for the cell. Their biogenesis relies on coordinated activity by various proteins to ensure structural integrity, selective transport, responses to stress, and virulence. In this study, we identified several proteins associated with these processes that show potential transcriptional regulation by the RNAP-σ⁵⁴ complex. The corresponding COGs are represented by green circles in Fig 6.

In the Myxococcia class, σ⁵⁴ regulation was enriched in COGs related to cell wall synthesis and maintenance, including lytic murein transglycosylases (COG0741), glycosyltransferases (COG0438), periplasmic polysaccharide export proteins (COG1596), UDP-3-O-acyl-N-acetylglucosamine deacetylases (COG0774), and membrane glycosyltransferases (COG2943). These COGs contribute to structural integrity, immune evasion, adhesion, and stress responses, supporting bacterial growth, adaptability, and pathogenicity. In the γ-Proteobacteria class, σ⁵⁴ regulation was enriched in COGs involved in cell wall synthesis and maintenance, including teichoic acid biosynthesis proteins (COG1922), pyridoxal phosphate-dependent enzymes (COG0399), UDP-N-acetylglucosamine 2-epimerase (COG0381), D-alanyl-D-alanine dipeptidase (COG2173), polysaccharide chain length determinant proteins (COG3765), and rod-binding proteins

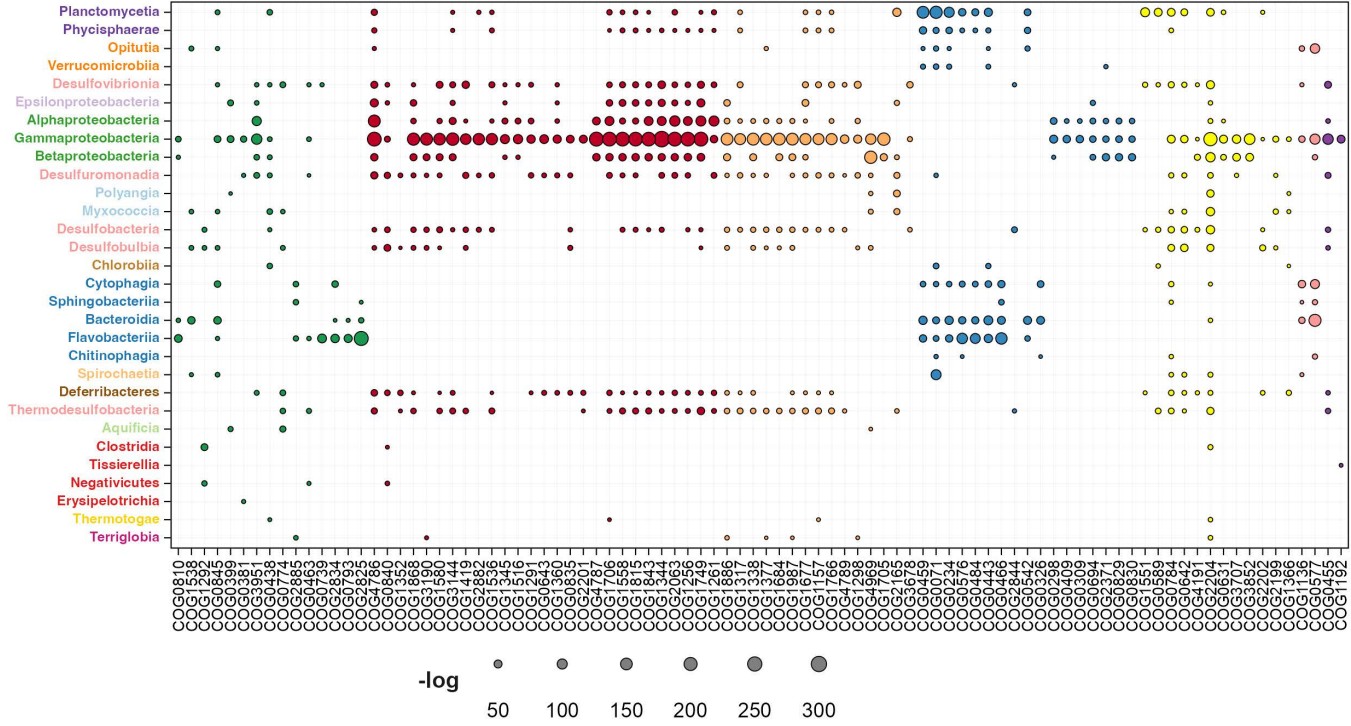

**Fig 6. Functional enrichment analysis of RNAP-σ⁵⁴-regulated genes in the cellular processes and signaling category.** Each circle represents a data point, with its diameter proportional to the negative log-transformed hypergeometric distribution (P value). Larger circles indicate greater enrichment for a specific COG subcategory, highlighting a stronger association with RNAP-σ⁵⁴ regulation. The figure's legend uses distinct colors to differentiate between functional groups for clarity. Green: cell wall/membrane/envelope biogenesis; red: cell motility. Orange: intracellular trafficking, secretion, and vesicular transport. Blue: posttranslational modification, protein turnover, and chaperones. Yellow: signal transduction mechanisms. Pink: defense mechanisms. Purple: cell cycle control, cell division, and chromosome partitioning.

(COG3951). These COGs play critical roles in cell wall integrity, morphology, antibiotic resistance, and environmental defense. Notably, COG3951 enrichment was also detected in α-Proteobacteria and Desulfuromonadia, indicating conserved functions across taxa.

Proteins linked to structural integrity and nutrient transport showed class-specific enrichment. In Flavobacteriia, the orthologous group COG0810, which includes TonB, was prominent. TonB, a periplasmic protein in Gram-negative bacteria, aids in importing nutrients and iron complexes by interacting with outer membrane receptors and energy-providing inner membrane complexes [76]. Although TonB's presence in Flavobacteriia is well-documented, its transcriptional regulation by the RNAP-σ⁵⁴ complex remains unconfirmed. In the Planctomycetia class, COG0668, linked to cell wall and membrane biogenesis, showed unexpected enrichment in RNAP-σ⁵⁴-dependent regulation. This group includes genes encoding mechanosensitive (MscS) channels, which help cells respond to mechanical stress by modulating ion flow. MscS channels are essential for osmoregulation, infection, and antibiotic resistance in pathogenic bacteria [23]. Given the ecological diversity of Planctomycetia, MscS channels may have evolved additional functions, although osmoregulation likely remains a key role due to their frequent exposure to osmotic stress. In addition, in *Salmonella Typhi*, *Escherichia coli*, and *Yersinia pestis*, members of the γ-Proteobacteria class, sequence homology analysis and predicted promoter sites have identified virulence-associated genes potentially regulated by σ⁵⁴ [63].

**Cell motility:** Cellular motility in bacteria, which includes swimming, swarming, and twitching, enables bacteria to explore their environment and respond to stimuli [12]. σ⁵⁴ regulates flagellar genes involved in flagellar synthesis and motor function, highlighting its importance in bacterial motility, as supported by previous studies [18,77] and findings in the current research. The regulatory influence of σ⁵⁴ on the expression of flagellar genes has been observed in human pathogens such as *Rhodobacter sphaeroides* [59], *Vibrio cholerae* [40,78], *Helicobacter pylori* [79], and *Campylobacter jejuni* [12], as well as in phytopathogens like *Xanthomonas oryzae* and *Ralstonia solanacearum* [17].

Fig 6 shows the distribution of genes related to cell motility (red circles) across bacterial phyla, with notable enrichment across various bacterial classes. These include a GTP-binding protein (COG1419) essential for energy transduction, motor switch proteins (COG1868 and COG1536) that regulate flagellar rotation, and motor components (COG1291 and COG1360) required for torque generation. Enrichment is most pronounced in γ-Proteobacteria, particularly in COG1419 and COG1868, followed by α- and β-Proteobacteria, Desulfobacteria, and Desulfuromonadia.

Patterns of enrichment in genes coding for flagellar basal body proteins underscore their critical role in motility across bacterial taxa. γ-Proteobacteria show the highest enrichment levels, particularly in proteins such as the basal body rod protein (COG4786), P-ring protein (COG1706), L-ring protein (COG2063), and FlaG (COG1334), which are vital for the structural integrity of the flagellar apparatus. Strong representation is also observed in α- and β-Proteobacteria, including the basal body components encoded by COG1815 and COG1558, highlighting their structural and functional conservation. Notably, the P-ring biosynthesis protein (COG1261) exhibits especially high enrichment in γ-Proteobacteria, emphasizing their specialization in basal body assembly. Enrichment patterns in Desulfobacteria, Desulfuromonadia, Planctomycetia, and Deferribacteres further underscore the remarkable conservation of these essential components across diverse bacterial lineages.

Our also results highlight the significant regulatory role of σ⁵⁴ in coordinating the synthesis of flagellar capping and hook-related proteins across bacterial classes, including α-, β-, and γ-Proteobacteria, as well as Desulfobacteria, Desulfovibrionia, and Thermodesulfobacteria. Within γ-Proteobacteria, the highest enrichment is observed in flagellin and hook-associated proteins essential for filament assembly and structural stability (COG1344) and the flagellar hook capping protein (COG1843). Additional enrichment in the flagellar hook protein FlgE (COG1749), the hook-length control protein (COG3144), and the hook-associated protein (COG1256) underscores their critical roles in hook assembly and length regulation.

Genes involved in flagellar assembly, particularly in β- and γ-Proteobacteria, show significant enrichment. These include the flagellin-specific chaperone FliS, which is essential for proper filament assembly (COG1516), as well as proteins associated with flagellar biosynthesis and the type III secretion system (COG3418 and COG2882).

Additionally, σ⁵⁴-dependent transcription was found to regulate motility to a lesser but still notable extent in other classes, including Desulfuromonadia, Desulfobacteria, Thermodesulfobacteria, Desulfovibrionia, and Desulfobulbia, all within the Thermodesulfobacteriota phylum; Opitutia within the Verrucomicrobiota phylum; Planctomycetia within the Planctomycetota phylum; and Deferribacteres within the Deferribacterota phylum.

Although the RNAP-σ⁵⁴ complex did not show statistically significant enrichment in regulating specific orthologous COG groups related to cell motility across certain bacterial phyla, a closer examination of individual organisms revealed σ⁵⁴-dependent promoters in motility-related genes. For example, in *Liquorilactobacillus nagelii* (Bacillota), including genes for flagellar motor switch phosphatase (COG1776), flagellar biosynthesis anti-sigma factor (COG2747), and flagellar hook-associated proteins (COG1256 and COG1344), and in Anoxybacillus flavithermus, including a gene encoding a flagellar hook (COG). More details are available in our repository at https://biocomputo.ibt.unam.mx/arpondb/.

**Intracellular trafficking/secretion and vesicular transport:** The RNAP-σ⁵⁴ complex plays a nuanced regulatory role in protein secretion systems. While its direct control over all genes in Type II (T2SS) and Type III (T3SS) secretion systems is not fully established, evidence indicates its involvement in specific genes across various bacterial species, including

*Campylobacter jejuni, Chlamydia trachomatis, Pseudomonas aeruginosa, Yersinia pseudotuberculosis, Erwinia amylov-
ora, Erwinia carotovora, and Pseudomonas syringae.*

Secretion systems like T2SS and T3SS are pivotal for bacterial pathogenesis, enabling host adhesion, barrier tra-
versal, immune evasion, and toxin production. Our findings support RNAP-σ54 complex's role in regulating pathogenicity-
associated genes in plant and animal/human pathogens. Enrichment of COGs linked to secretion systems was observed
across several bacterial classes, including α, β, and γ -Proteobacteria, Desulfobacteria, Planctomycetia, Polyangia, and
Myxococcia, as illustrated by orange circles in Fig 6.

The T2SS facilitates intracellular trafficking and vesicular transport and are composed by different protein components
(COG2165, COG3031, COG4795, COG1450, COG1459, COG2804, COG3156, and COG4972) which are predominantly
enriched in Myxococcia. Lesser enrichment is noted in Polyangia and Planctomycetia for COG2165.

The T3SS primarily functions to deliver effector proteins directly into host cells. Our study identified statistically sig-
nificant enrichment of genes associated with this secretion system (COG1886, COG1766, COG4789, COG1317) pre-
dominantly in γ-Proteobacteria, with lower levels of enrichment observed in α-Proteobacteria, β-Proteobacteria, and
Desulfobacteria.

Genes related to flagellar biosynthesis and pilus assembly, highlight σ54's role in motility and interactions with host cells.
For instance, Tfp Pilus Assembly Protein PilA (COG4969), strongly enriched in β-Proteobacteria, with γ-Proteobacteria
following. Flagellar Components, FliR (COG1684), FlhB (COG1377), and FliQ (COG1987) exhibit peak enrichment in
γ-Proteobacteria, with moderate presence in Desulfobacteria, β, and γ-Proteobacteria.

**Posttranslational modification/protein turnover and chaperones:** Chaperones play a vital role in protein folding, sta-
bilization, transport, and degradation, contributing to the maintenance of cellular homeostasis. In Fig 6 (blue circles), two
key protein clusters are highlighted within the posttranslational modification, protein turnover, and chaperone category.

The first cluster includes heat shock proteins such as the molecular chaperonin GroEL (COG0459), the small heat
shock protein IbpA (COG0071), the co-chaperonin GroES (COG0234), the nucleotide exchange factor GrpE (COG0576),
and the chaperone complex proteins DnaJ (COG0484) and DnaK (COG0443). These proteins operate in a coordinated
manner: DnaJ initially binds to misfolded proteins, transferring them to DnaK, which stabilizes and assists in refolding
substrates through ATP-driven conformational changes. The substrates are subsequently handed off to the GroEL-GroES
chaperonin system for final folding, a process facilitated by GrpE [80].

This cluster is significantly enriched in the Planctomycetota and Bacteroidota phyla, with the highest representation
in the Flavobacteriia class of Bacteroidota, followed by Bacteroidia and Cytophagia. In Planctomycetota, enrichment is
concentrated in Planctomycetia, with a smaller presence in Phycisphaerae. Notably, the transcription of GrpE in Bacteroi-
des thetaiotaomicron and Flavobacterium psychrophilum, as well as GroES in *B. thetaiotaomicron*, is σ54-dependent [80].
Additionally, COG0326, part of the HSP90 family, is enriched in the Bacteroidia class. Elements linked to protein turnover
were also identified within these phyla, including ATP-dependent proteases COG0542, COG0466, and COG1066. These
proteases are enriched in the Flavobacteriia and Bacteroidia classes of Bacteroidota and in the Planctomycetia class of
Planctomycetota.

The second cluster includes proteins involved in nitrogenase and hydrogenase enzyme assembly, oxidative stress
response, and degradation of misfolded proteins. Key members of this cluster are NifU (COG0694), hydrogenase matu-
ration factors (COG0298, COG0409, COG0309), proteasome-type protease (COG3484), and peroxiredoxin (COG0450),
with significant enrichment observed in α, β, and γ-Proteobacteria. NifU assembles Fe-S clusters, which are essential for
electron transfer, enzyme catalysis, and regulation. Hydrogenase maturation factors, such as HypC/HupF, assist in hydro-
genase enzyme maturation by incorporating metal cofactors like Fe-S clusters. Proteasome-type proteases ensure protein
quality by degrading misfolded proteins, thus maintaining cellular homeostasis. Notably, a chaperone-specific cluster
enriched in Spirochaetia includes proteases with potential chaperone activity for protein degradation (COG0533).

**Signal transduction mechanisms:** This subcategory within *Cellular processes and signaling* comprises COGs associated with proteins that are crucial for bacterial signal transduction pathways. These COGs are represented by yellow circles in Fig 6, particularly within two-component systems (TCSs). TCSs consist of a sensor histidine kinase (HK) and a response regulator (RR), enabling bacteria to adapt to environmental stimuli and regulate virulence factors. These systems detect environmental changes and trigger signal transduction cascades that activate or repress target genes, including those involved in stress responses [81].

Notable COGs overrepresented in this subcategory include those related to stress response (COG0589), transcriptional regulation (COG2204, COG3707, and COG3852), and signal transduction (COG4191). For instance, COG1551, associated with carbon storage regulation, intersects with quorum sensing and motility, reflecting its broader regulatory roles. Prominent mechanisms also include histidine kinases (COG0642, COG5000) and CheY-like receivers (COG0784), underscoring a dynamic regulatory network across bacterial classes such as Planctomycetia, Desulfobacteria, and Gammaproteobacteria.

In Planctomycetia, notable enrichment of COG0589 includes proteins such as the UspA domain protein, T2SS protein E, and putative universal stress proteins. UspA proteins help bacteria adapt to nutrient scarcity, oxidative stress, and toxins by integrating environmental signals into cellular responses [82]. Protein E, part of the T2SS system, plays a role in secretion and may be regulated in response to stress signals [83].

In β- and γ-Proteobacteria, COG3852 includes histidine kinases like NtrB, key for nitrogen fixation. The process is regulated by a two-component system, with NtrB detecting nitrogen availability and activating NtrC. NtrC, containing PAS-type domains, senses environmental signals (e.g., nitrogen levels) and undergoes conformational changes. This activates NtrB, which phosphorylates NtrC. The phosphorylated NtrC (NtrC-P) then binds DNA promoters, activating transcription of genes required for nitrogen fixation [33]. Furthermore, our method identified promoters not only in NtrB but also in homologous NtrC genes within COG2204, a cluster of DNA-binding transcriptional response regulators. These regulators, part of the NtrC family, are characterized by REC, AAA-type ATPase, and Fis-type DNA-binding domains, and also include the transcriptional regulator ZraR. The analysis revealed significant enrichment of 414 genes in the COG2204 cluster with potential $\sigma^{54}$-dependent promoters, notably in Pseudomonadota and other phyla like Planctomycetota and Thermodesulfobacteriota.

In Pseudomonadota, nitrogen fixation is regulated by the NtrB/NtrC two-component system, but its conservation and functional presence in Planctomycetota, Thermodesulfobacteriota, and Spirochaeotota are limited. In Planctomycetota, the predicted genes correspond to ZraR, a response regulator in the ZraS/ZraR two-component system, involved in stress responses to zinc and heavy metals. ZraR, activated by ZraS, modulates gene expression, often working with $\sigma^{54}$ to initiate transcription in response to environmental signals [84]. It can also autoregulate the operon encoding ZraS and ZraR, contributing to feedback regulation.

Another $\sigma^{54}$-dependent regulated COG related to nitrogen metabolism is COG3707, which groups a cluster of genes encoding proteins with AmiR and NasR Transcription Antitermination Regulator (ANTAR) domains, RNA-binding domains involved in transcription antitermination. These domains are found in response regulators of two-component systems and single-component sensory regulators. The analysis predicts 66 $\sigma^{54}$-dependent promoters in proteins grouped in this COG, primarily in β and γ-Proteobacteria. Notably, the COG includes NasT, which regulates the *nasAB* operon, responsible for nitrate and nitrite reductases, by antitermination through binding to stem–loop structures in the operon's leader region.

Our analysis showed significant enrichment of COG4191, which includes various response regulators, particularly in Desulfobacteria. This COG groups proteins involved in signal transduction and C4-dicarboxylate transport, primarily those signaling through histidine kinases. C4-dicarboxylate transport sensor proteins regulate the uptake of C4-dicarboxylates, such as fumarate, succinate, malate, and aspartate, which are essential for bacterial metabolism in specific environmental conditions. These bacteria use C4-dicarboxylates as carbon sources and electron acceptors, enabling them to detect and transport these compounds.

There was notable enrichment of COG1551 within the Planctomycetia class, which includes proteins associated with the carbon storage regulator (CsrA). CsrA plays a key role in bacterial metabolic regulation, swarming behavior, and quorum sensing. These RNA-binding proteins manage carbon storage and allocation, influencing bacterial motility and community behavior. They regulate gene expression post-transcriptionally by modulating mRNA stability and translation involved in carbon metabolism. The csrA gene has been shown to have σ54-dependent promoters in *Pirellula* species strain 1, a member of Planctomycetota, where the σ54 regulon has been studied [27].

The presence of RNAP-σ54-dependent promoters in genes such as *ntrC-ntrB*, *ntrY-ntrX, nasT*, and other EBPs and their sensor kinases suggests coordinated regulation in response to environmental conditions. This regulation ensures the simultaneous production of sensors (such as NtrB) and response regulators (NtrC), enabling the proper detection and transduction of signals. The functional specialization of these EBPs and sensor kinases allows bacteria to adapt to various environmental conditions.

**Defense mechanisms:** COGs exhibiting significant enrichment of σ54-mediated transcription in this category are represented by pink circles in Fig 6. These include COG0577, which corresponds to the permease component of the ABC-type antimicrobial peptide transport system. Notably, this COG was enriched in the γ-Proteobacteria, Opitutia, and Bacteroidia classes within the Pseudomonadota, Verrucomicrobiota, and Bacteroidota phyla. Permease components are crucial for transporting various molecules across the cell membrane and display considerable structural diversity and substrate specificity [85–87]. This adaptation is especially relevant for pathogens like *Pseudomonas aeruginosa*, which use ABC transporters to evade host immune defenses [88–90].

In Opitutia, we observed enrichment of COG1136, which includes genes encoding the ATPase component of ABC-type antimicrobial peptide transport systems. These ATPases hydrolyze ATP to power substrate transport, crucial for bacterial survival. In soil-dwelling bacteria like Opitutia, these systems also aid in nutrient acquisition and biofilm formation, showcasing their functional versatility beyond antimicrobial defense.

COG0842, enriched in Bacteroidia, represents the permease component of ABC-type multidrug transport systems, key to bacterial multidrug resistance (MDR). These permeases transport various substrates across membranes, and their upregulation in response to environmental stresses or antibiotics boosts bacterial survival, especially in gut microbiota exposed to xenobiotics and antibiotics.

The FtsX-like permease family, regulated by the RNAP-σ54 complex, plays a crucial role in maintaining cell envelope integrity, preventing membrane damage, and preserving cell viability under antimicrobial stress by regulating the influx or efflux of antimicrobial peptides (AMPs). This study provides the first evidence linking RNAP-σ54-mediated regulation to this family of ABC-type systems, revealing a previously unrecognized regulatory axis.

The differential enrichment of COGs across bacterial classes reflects ecological and evolutionary adaptations. In Opitutia, the presence of permease (COG0577) and ATPase (COG1136) components suggests a strong antimicrobial defense system, likely crucial for survival in competitive environments. In Bacteroidia, the prominence of multidrug transport systems (COG0842) aligns with their role in gut microbiota, where exposure to various xenobiotics and antibiotics is common.

**Cell cycle control/cell division and chromosome partitioning:** In this subcategory, COG1192 and COG0455, enriched in γ-Proteobacteria, Desulfobacteria, and Desulfuromonadia, cluster proteins related to ATPase activity, crucial for energy-driven processes. These enrichments are visually represented by purple circles in Fig 6. Notably, ParA/MinD proteins, involved in chromosome segregation during cell division, form a dynamic gradient in the bacterial cell, aiding in the movement of replicated chromosomes to opposite poles [91]. These proteins interact with FtsZ to coordinate chromosome segregation and septum formation [92]

**Information storage and processing.** In our analysis, the third functional category concerned genes involved in fundamental processes associated with DNA replication, repair, recombination, chromatin remodeling, transcription, ribosome biogenesis, and genes responsible for maintaining and utilizing the genetic material within cells. Fig 7 illustrates the enrichment of genes within this category that could be transcriptionally regulated by σ54.

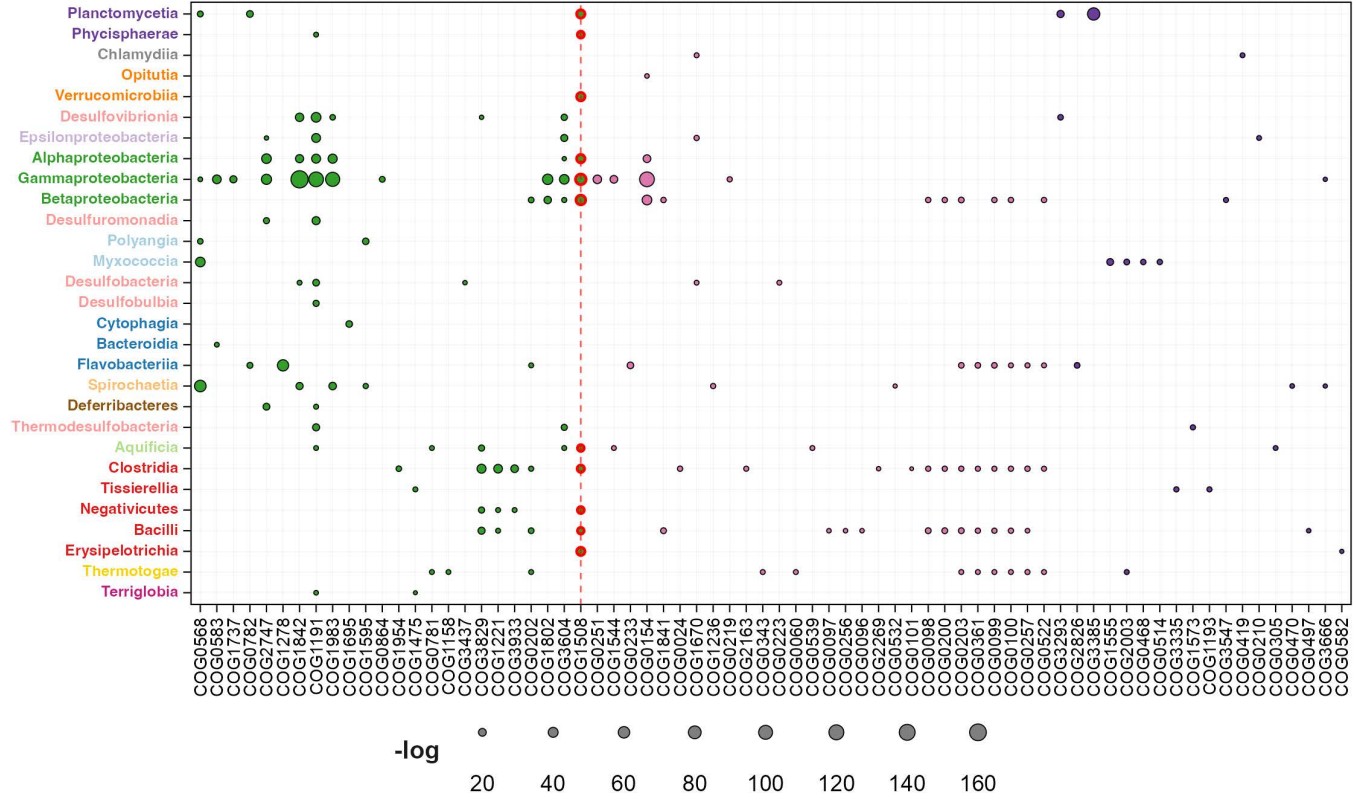

**Fig 7. Functional enrichment analysis of RNAP-σ⁵⁴-regulated genes in the Information Storage and Processing category.** This figure highlights functional enrichment analysis results for RNAP-σ⁵⁴-regulated genes within the *Information Storage and Processing* category of the COG database. Each circle represents a data point, with its size proportional to the negative log-transformed hypergeometric P value, signifying enrichment levels. Larger circles indicate stronger associations with RNAP-σ⁵⁴ regulation. A color-coded legend distinguishes COG subcategories, while the red dashed line marks COG1508, representing the orthologous group of genes encoding the σ⁵⁴ or RpoN sigma factor. Functional groups are represented using distinct colors in the legend to facilitate interpretation. Green: transcription. Pink: translation, ribosomal structure, and biogenesis. Purple: recombination and repair.

**Translation/ribosomal structure and biogenesis:** The first subcategory within the *Information Storage and Processing* category, which focuses on the *translation/ribosomal structure and biogenesis* group and is represented by pink circles in Fig 7. Many of the orthologous clusters identified as significantly regulated by RNAP-σ⁵⁴ complex were associated with ribosomal proteins, including L17, S8, L6P/L9E, S5, L15, L18, S14, S11, L24, L36, S4, and L30/L7E (COG0203, COG0096, COG0097, COG0098, COG0200, COG0256, COG0199, COG0100, COG0198, COG0257, COG0522, and COG1841, respectively). This enrichment of RNAP-σ⁵⁴ complex transcriptional regulation of genes encoding ribosomal proteins is predominantly observed within the Flavobacteria phylum, with significant enrichment also in ε, α and β-proteobacteria. Additionally, we observed significant enrichment of RNAP-σ⁵⁴ complex regulation of genes encoding the Asp-tRNA_Asn/Glu-tRNA_Gln amidotransferase (COG0154) in the α, β, and γ-Proteobacteria classes, along with translation initiation factor 1 (IF-1) (COG0361) within the Flavobacteria and ε-Proteobacteria classes.

**Transcription:** Within the transcription subcategory, a distinct pattern emerges in the regulation of the RNAP-σ⁵⁴ complex. This pattern includes various specialized sigma subunits of DNA-directed RNA polymerase, which are represented by green circles in Fig 7. Notably, the COG1191 subtype predominates, particularly among α, and γ-Proteobacteria. Similarly, other subsets of specialized DNA-directed RNA polymerase sigma subunits, such as those encoding σ⁵⁴ (COG1508) or σ⁷⁰/σ³² (COG0568), exhibited a consistent association with σ⁵⁴ regulation. Furthermore, the enrichment of genes related to

σ⁵⁴ regulation included genes responsible for transcriptional regulation (COG1802, COG1221, COG1737, COG1802 and COG1392), transcriptional anti-terminators (COG3933, COG2747 and COG1954), the DNA-directed RNA polymerase α subunit/40 kD subunit (COG0202), and cold shock proteins (COG1278) and phage shock protein A (COG1842).

In specific bacterial classes, such as Bacilli, Clostridia, Erysipelotrichia, Verrucomicrobiia, Plantomycetia, α, β, and γ-Proteobacteria, a notable enrichment of genes encoding σ⁵⁴ (RpoN sigma factor, COG1508) and transcribed via σ⁵⁴ promoters was observed, implying autoregulation of the transcription of this sigma factor. While this regulatory phenomenon is not significant in other phyla, it has been documented in select organisms such as *Geobacter sulfurreducens, δ*-Proteobacterium [28], and Dokdonia sp. MED134, a member of the Bacteroidetes phylum, where the *rpoN* gene is part of the σ⁵⁴ regulon [93].

**Replication/recombination and repair:** Fig 8 highlights the subcategory of replication, recombination, and repair COGs, represented by purple circles. Notably, we observe enrichment of DNA repair proteins within COGs such as COG2003, COG125, and COG119. These COGs play a critical role in repairing DNA damage and preserving genomic integrity. Their consistently low E-value across diverse bacterial phyla underscores their evolutionary conservation and importance. By maintaining genome stability, these proteins enable bacteria to endure environmental stresses that could otherwise compromise DNA integrity. This universal conservation across bacterial phyla suggests that these repair mechanisms are fundamental for bacterial survival and evolutionary success.

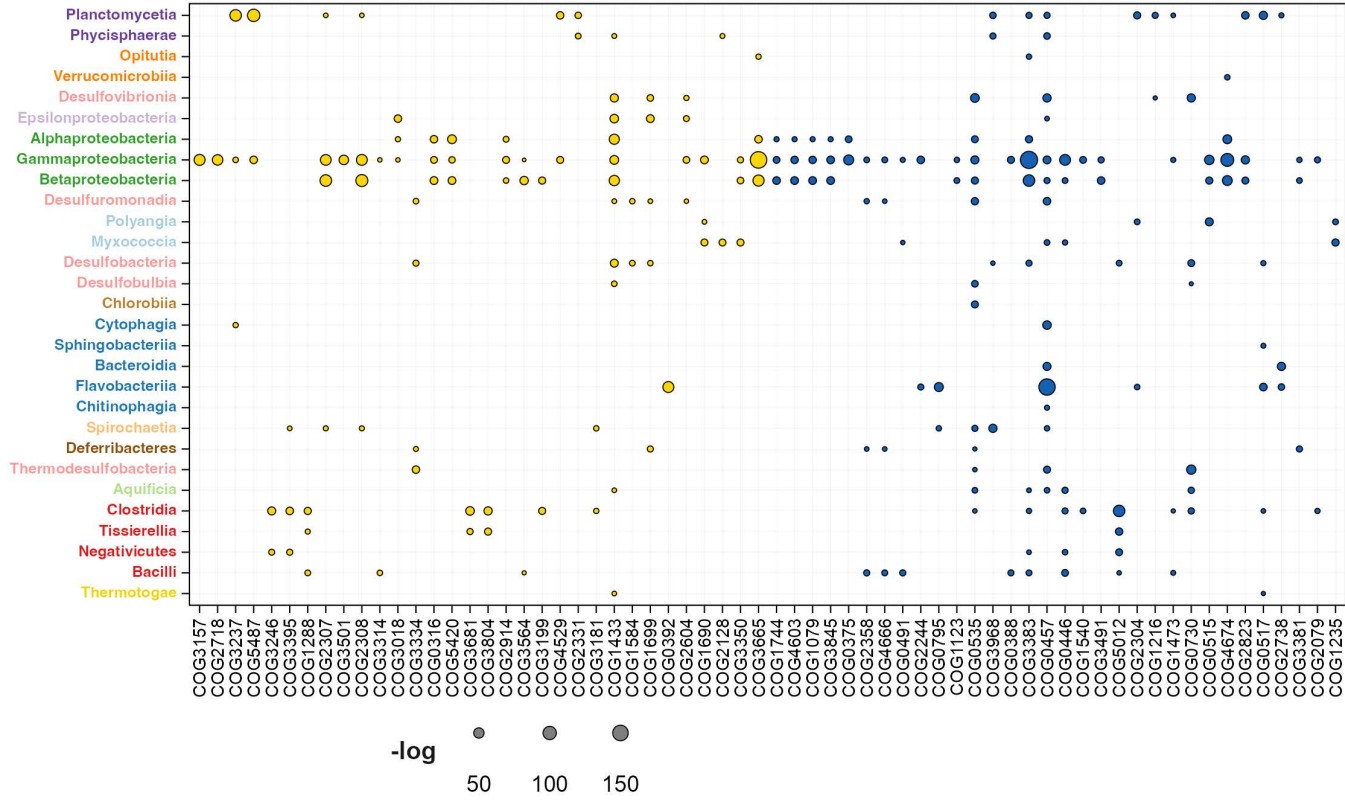

**Fig 8. Functional enrichment analysis of RNAP-σ⁵⁴-regulated genes in the Poorly characterized category.** This figure shows the results of a functional enrichment analysis focusing on genes regulated by the RNA polymerase σ⁵⁴ within the *poorly characterized* category in the COG database. Each data point is represented by a circle, with the circle's diameter directly proportional to the negative log-transformed hypergeometric distribution (P value) value. Larger circles correspond to a greater degree of enrichment for a particular COG subcategory, indicating a significant association with RNAP-σ⁵⁴ regulation. The legend accompanying the figure indicates the unique color that is assigned to each COG subcategory depicted on the graph to allow for clear differentiation of functional groups. Blue: general function prediction only. Yellow: function unknown.

In addition, our analysis identifies three COGs associated with transposases and related proteins (COG3666, COG3464, and COG3385). These proteins are crucial for DNA rearrangement and the activity of mobile genetic elements, driving horizontal gene transfer and genetic diversity. Such mechanisms significantly enhance bacterial adaptability and evolution. Transposases exhibit notable enrichment in taxa like Spirochaetia and Bacteroidia, where low E-values reflect their functional importance. Interestingly, COG3385 is significantly enriched in the Planctomycetia class. However, caution is warranted when interpreting these results due to the repetitive nature of transposons, which can obscure specific regulatory associations, such as those mediated by RNAP-$\sigma^{54}$ complex.

Finally, we identify COGs associated with ATPases involved in DNA replication and repair, including COG126, COG125, and COG127. These ATPases are essential for processes such as DNA unwinding during replication and facilitating repair mechanisms. They are particularly prominent in Spirochaetia and Myxococcia, where low E-values underscore their significant roles. Their conservation highlights their importance in ensuring bacterial genome stability and supporting core cellular functions.

**Poorly characterized.** The term *poorly characterized* in the COG database refers to a group of proteins that exhibit sequence conservation across diverse organisms. These proteins, despite their conservation, have not been thoroughly studied or characterized, leading to a lack of comprehensive functional annotation or detailed experimental data. However, their potential significance cannot be overstated. Although these proteins are believed to possess conserved functions based on their sequence similarity, their precise roles or functions remain largely unknown. The COG database considers two different classes of proteins within this category—*function unknown* and *general function prediction only*. The relative enrichment of $\sigma^{54}$ regulation in these groups is presented in Fig 8.

**General function prediction only:** The *general function prediction only* category refers to those COGs for which limited information is available regarding the functions of their protein members; these are represented in Fig 8 by blue circles. For example, COG3383 is annotated as "uncharacterized anaerobic dehydrogenase"; however, from a closer look at the individual description of the 176 members regulated by $\sigma^{54}$ in this COG, two different classes of enzymes emerge within this COG: 42% correspond to nitrate reductase enzymes, while 32% correspond to formate dehydrogenases. The transcriptional regulation of this COG by the RNAP-$\sigma^{54}$ complex is enriched in α, β, and γ-Proteobacteria and Bacilli. COG0402 is referred to in the COG database as "cytosine deaminase and related metal-dependent hydrolases". Nevertheless, a detailed analysis of the gene descriptions revealed that these COG clusters cytosine deaminase (42%) and guanine deaminase (32%) enzymes. Members of COG3383 and COG0402 are regulated by $\sigma^{54}$ in the α, β, and γ-Proteobacteria classes.

Others COGs such as COG1744, COG460, COG1079, COG3845, etc. refer to predicted orthologous genes with shared ancestry and potentially similar functions. These are annotated as poorly characterized in *general function prediction only*, but suggest they play roles in ABC transport systems, indicating potential but unclear roles in cellular processes. They have a significant enrichment across different classes of Proteobacteria (in α, β, and γ-Proteobacteria).

The Planctomycetes phylum exhibits a fascinating diversity in its COGs, many of which are linked to *poorly characterized* proteins with intriguing potential. A number of these COGs, such as COG0457 (TPR repeat) and COG0517 (CBS domain), are involved in general function predictions, hinting at novel mechanisms of protein-protein interactions and cellular regulation that remain to be fully explored. Many of these COGs are associated with membrane transport and periplasmic proteins, such as predicted permeases and lipoproteins, suggesting a complex network of nutrient acquisition, particularly in nutrient-poor environments. Additionally, some COGs predict proteins related to nitrogen metabolism, including those akin to glutamine synthetase and glycosyltransferases, which may contribute to Planctomycetes' roles in nitrogen cycling, particularly in wastewater treatment and symbiotic relationships. The high number of uncharacterized proteins reflects the unique and still-mysterious roles these organisms play in their ecological niches, reinforcing their potential for biotechnological applications, such as bioremediation and symbiotic interactions with plants or other microorganisms. These findings highlight Planctomycetes' evolutionary novelty and adaptability, pointing to their critical roles in environmental processes and their potential for future discovery in microbiology.

**Function unknown:** The COGs categorized under *function unknown* represent genes without associated information. However, our findings suggest potential coordinated regulation of their transcription by σ54 and are represented by yellow circles in Fig 8. Despite being labeled as an *uncharacterized conserved protein*, a closer examination of the functional definitions of these genes reveals that, for specific genomes, more precise knowledge exists about their functions. For instance, out of the 102 genes regulated by σ54 in α, β, and γ-Proteobacteria within COG3665, 59% were identified as "urea carboxylase." Similarly, despite COG3501 being referred to as an *uncharacterized conserved protein* in the COG database, we determined that 72% of its members are associated with the "type VI secretion protein VgrG," a structural component that is crucial for the assembly of the T6SS apparatus and is primarily regulated by the RNAP-σ54 complex in the γ-Proteobacteria class. Proteins clustered in COG1288, COG2170, and COG3221 are predicted to be associated with membranes, which implies involvement in processes like transport, signaling, or interaction with the environment. These predictions are often made based on hydrophobic domain analyses.

Other groups, such as proteins with SCP/PR1 domains (COG2340) in Planctomycetes and proteins with CXXC pairs (COG3880) in Opitutia, offer potential functional insights. SCP/PR1 domains are linked to pathogenesis and may also possess enzymatic functions in certain contexts [94]. These findings indicate that uncharacterized proteins are not simply evolutionary remnants but could represent essential components of bacterial physiology, awaiting further functional characterization.

### Online access to our data and analyses through the aRpoN-DB Web Page

The aRpoN-DB Web Page is a web-based platform designed to store, visualize, and explore data derived from a comprehensive comparative analysis of σ54-dependent regulons across multiple bacterial phyla. This freely accessible platform, available at https://biocomputo.ibt.unam.mx/arpondb/, offers an intuitive interface and interactive tools that provide researchers with valuable insights into the cellular functions regulated by σ54.

The main page presents global analyses through interactive graphs and tables, including a summary of bacterial classes studied. Each class links to a list of organisms organized by genus, which in turn redirects to detailed organism-specific pages. These pages display putative σ54-dependent regulons, including information such as predicted promoter sequences, cognate DNA motifs, statistical enrichment values (E-values), COG-based functional classifications, and downstream operon genes with functional annotations. All data is downloadable in CSV format for custom analyses.

The platform provides visualizations covering five key aspects of σ54-regulated pathways: (1) the phylogenetic distribution of *rpoN* genes; enrichment analyses of the RNAP-σ54 regulon in (2) metabolism-related genes, (3) Cellular processes and signaling related genes, (4) Information storage and processing related genes, and (5) genes with poorly characterized functions. Additionally, a curated set of 33 logos representing conserved DNA motifs in promoter sequences, categorized by taxonomic class, is available in an interactive grid format for download, supporting further research into gene regulation.

### Distribution of COG functional categories across phylogenetic classes

To summarize our main findings on the diverse regulatory roles of σ54 across a wide range of bacterial lineages, we present the distribution of COG functional categories associated with σ54-regulated genes across different phylogenetic classes (Fig 9). This overview not only validates earlier reports on the role of σ54 in regulating specific metabolic and cellular processes within certain phylogenetic groups but also highlights previously unrecognized taxa where σ54 may play important regulatory roles.

### Conclusion

Nearly three decades have passed since the functional characterization of the product encoded by the *rpoN* gene as an RNA polymerase sigma factor [95]. Over this time span, the role of σ54 in various cellular processes has been extensively

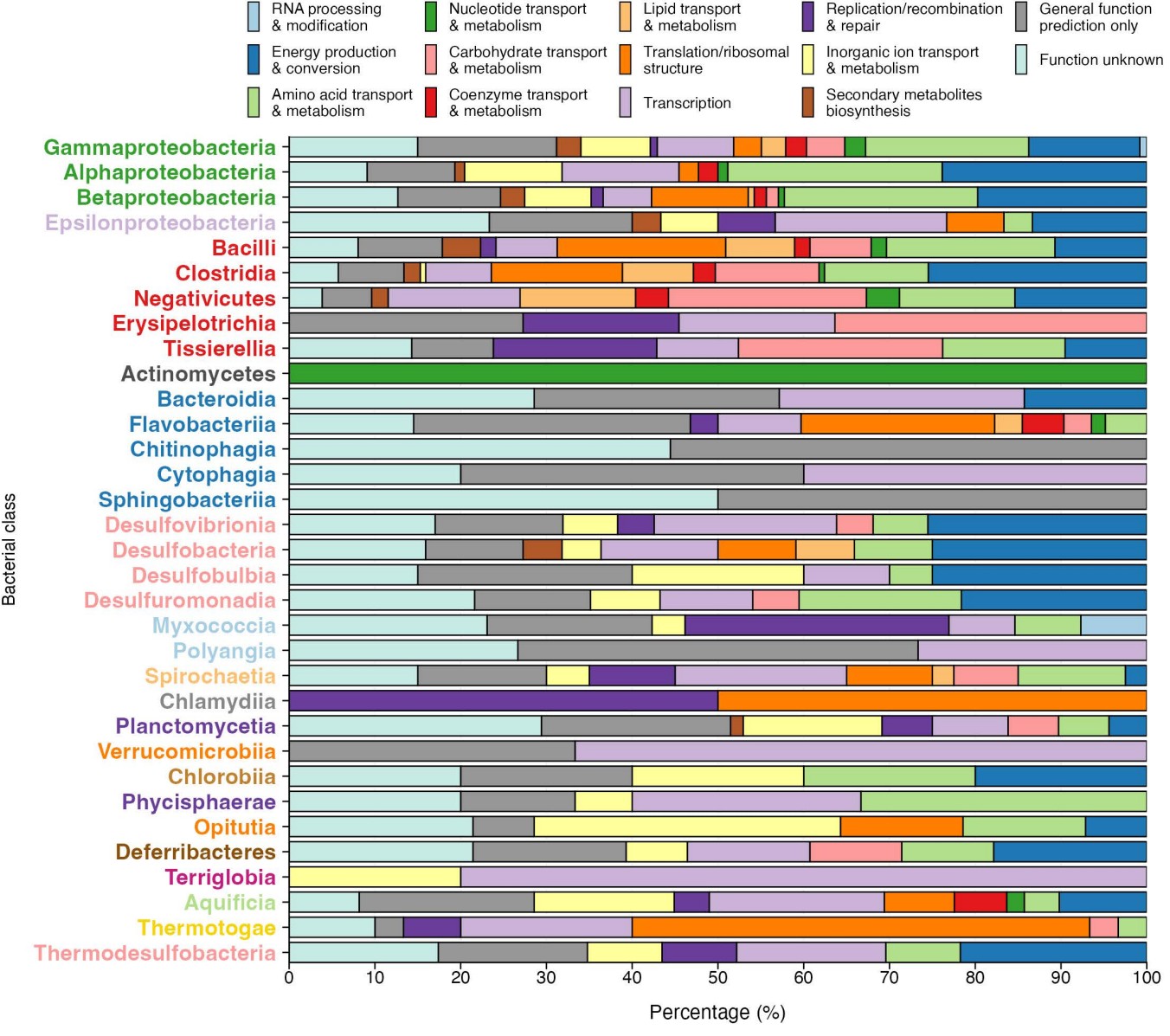

**Fig 9. COG functional categories among phylogenetic classes.** σ54 regulated genes were first grouped in relation to their respective COG orthologous groups, which were then subcategorized into broader functional classes. The relative abundance of each functional category within each phylogenetic class is shown as percentage values.

documented. These processes include nitrogen metabolism, flagellar biosynthesis and motility, urease activity formation, flagellar biosynthesis and motility, stress response pathways, virulence factors, and carbon metabolism. The role of σ54 in regulating these and other cellular functions has been documented and reviewed, with a focus on specific cellular processes or phylogenetic groups.

In our study, we exploited the diverse and vast set of available bacterial genome sequences and developed a bioinformatic approach to elucidate the regulatory network of σ54 across various phylogenetic groups. Our approach was initially based on the construction of a position-specific scoring matrix (PSSM) derived from the regulatory regions of

720 experimentally determined *rpoN* genes of 56 organisms from different phylogenetic origins. Afterwards, we implemented an iterative motif discovery and refinement process, combining MEME and MAST within each of our 33 phylogenetic classes across 16 distinct bacterial phyla. Through rigorous statistical analysis, we identified the most significantly enriched group of orthologous genes transcribed by σ54 within each major taxonomic class to provide novel insights into the regulatory landscape governed by σ54 across diverse bacterial lineages.

As anticipated, our bioinformatic approach led to the successful identification of all the cellular functions previously documented to be transcriptionally regulated by σ54 within the major phylogenetic groups. Furthermore, our analysis revealed the full extent of the σ54 regulon in additional groups in which previous information was lacking, thereby providing a statistical assessment of the prevalence of σ54 regulation within each phylogenetic clade. In addition, our study revealed unreported cellular processes that were regulated by σ54 in particular phylogenetic clades, such as the ABC-type antimicrobial peptide transport system used as a defense mechanism in the γ-Proteobacteria and Bacteroidia classes, the genes encoding ribosomal proteins within the Flavobacteria phylum, or the genes involved in the chromosome partitioning process, a process essential for precise cell cycle control and division, in Thermodesulfobacteria and the α, ∂, and γ-Proteobacteria classes.

As extensively documented, EBPs play an essential role in σ54-dependent transcription by facilitating the transition from the stable closed complex of the DNA-bound RNA polymerase-σ54 holoenzyme to an open complex to initiate the transcription process. Through a comprehensive sequence analysis involving over 1,400 nonredundant genomes, we demonstrated a clear correlation between the presence of these EBPs in bacterial genomes and that of one of the σ54 coding genes. This correlation underscores a coevolutionary process involving these regulatory elements.

Although our inference regarding σ54 regulons across different phyla relies solely on sequence analysis, the accuracy of our bioinformatics prediction model was significant when benchmarked against experimentally defined genes transcribed by σ54 in *E. coli* that were documented in the EcoCyc database. Furthermore, our predictions align well with the established understanding of the primary cellular processes that are regulated by σ54, which has been extensively documented. This finding elucidates the differential prevalence of σ54 regulation among the main bacterial phyla because of the selective processes throughout evolution that have shaped the regulatory landscape of this sigma factor, which is of unique importance in the regulation of genes in bacteria.

## Supporting information

**S1 Fig. Phylogenetic analysis based on 16S rRNA sequences shows that *Arthrobacter citreus* NEB 577 aligns more closely with the Bacillota phyla than with Actinomycetota.** The 16S rRNA sequences from five representative organisms within each of the 33 phylogenetic classes included in our study were aligned and used to construct the phylogenetic tree shown in the figure. This tree reveals that *Arthrobacter citreus* NEB 577 clusters more closely with members of the Bacillota phylum (red) than with those of the Actinomycetota (yellow).
(TIF)

**S2 Fig. Distribution of COGs associated with σ54-regulated genes across phylogenetic classes in the metabolism category.** The bar chart illustrates the number of COGs per phylogenetic class, organized by their functional descriptions. *Metabolism*-related COGs are further subdivided into functional subcategories to highlighting enriched classes. The COGs included in each category were selected based on their statistical enrichment values, as determined by hypergeometric distribution statistics, as detailed in Fig 5.
(TIF)

**S3 Fig. Distribution of COGs associated with σ54-regulated genes across phylogenetic classes in the Cellular processes and signaling category.** The bar chart illustrates the number of COGs per phylogenetic class, organized by their functional descriptions. *Cellular processes and signaling*-related COGs are further subdivided into functional

subcategories to highlighting enriched classes. The COGs included in each category were selected based on their statistical enrichment values, as determined by hypergeometric distribution statistics, as detailed in Fig 6.
(TIF)

**S1 Table. Study organisms from 33 classes from 16 prokaryotic phyla.** This table displays organisms grouped by their class and phylum, with one representative organism per genus. Organisms were organized according to their respective phylogenetic classes across 16 prokaryotic phyla. The "Class" column categorizes each organism based on its phylogenetic classification. Only classes containing at least four organisms encoding the *rpoN* gene were included, refining the dataset to a more focused and relevant group. The final dataset comprises 1,414 organisms, offering a comprehensive and representative sample of prokaryotic diversity. Additionally, model organisms such as *Escherichia coli* (eco) and *Bacillus subtilis* (bsu) were included as reference points for comparative analysis. Each organism is identified by a three-letter code derived from the KEGG GENOME Database.
(XLS)

**S2 Table. Initial set of seed RpoN-dependent promoter sequences used to generate the seed MEME-derived position-specific scoring matrix (PSSM).** 720 experimentally validated $\sigma^{54}$-dependent promoter sequences derived from 56 bacterial species were compiled from major regulatory databases, including RegPrecise 3.0 [49], DBTBS [50], and EcoCyc [51], as well as from the comprehensive compilation by Barrios et al. (1999) [52].
(XLSX)

**S3 Table. Seed position-specific scoring matrix (PSSM).** The initial set of seed 720 experimentally validated $\sigma^{54}$-dependent promoter sequences were used to generate the seed MEME-derived position-specific scoring matrix (PSSM).
(TXT)

**S4 Table. List of organisms used as a negative control in our $\sigma^{54}$-specific EBP search.** 380 genome sequences form Archaea and 36 Eukaryota, were used as negative control to identify $\sigma^{54}$-specific EBP proteins.
(XLSX)

**S5 Table. Distribution of Organisms with Multiple *rpoN* Genes by Phylogenetic Class.** This table presents the organisms in the studied dataset that harbor more than one *rpoN* gene. They have been grouped by their respective phylogenetic classes. Each organism is identified by a three-letter code derived from the KEGG genome database. Additionally, the table provides the number of organisms in each class that contain the *rpoN* gene alongside the total number of organisms per class.
(XLS)

**S6 Table. RpoN sequence similarities.** The amino acid sequences of RpoN proteins from model organisms and Actinomycetota were compared, and the percentage similarities are summarized in separate tables for the AID, CBD, and DBD domains.
(XLSX)

**S7 Table. Pfam scores for the AID, CBD, and DBD domains of RpoN proteins from Actinomycetota species.** The amino acid sequences of RpoN proteins from both model organisms and Actinomycetota species were analyzed using the *hmmsearch* program to identify their corresponding conserved domains. The Pfam scores for these domains are presented in the table.
(XLSX)

**S8 Table. Distribution of EBPs across prokaryotic classes.** He proteomes of the organisms included in our study were analyzed to identify enhancer-binding proteins (EBPs) using a pipeline that combines domain-based detection using

the *hmmsearch* program and the Pfam Sigma54_activat motif (PF00158), with domain architecture analysis using MAST, as described in the Methods section.
(XLSX)

**S9 Table. Architectural arrangements of Pfam domains in RpoN proteins across phylogenetic classes.** This table details the diverse architectural arrangements of Pfam domains within RpoN proteins, identifying the PF00309 (AID), PF04552 (DBD), and PF04963 (CBD) for domain analysis. The architectural arrangements are categorized and labeled as AID-DBD-CBD, AID-CBD, AID-DBD, and CBD-DBD, highlighting the distribution of each RpoN protein arrangement across different classes.
(XLSX)

**S10 Table. Sequence variations of false negative σ54-promoters of our study.** Bona fide promoter sequences that were not detected by our computational protocol were analyzed to identify deviations from the consensus promoter sequence.
(XLSX)

## Acknowledgments

We sincerely thank Shirley Ainsworth for her bibliographical services, Arturo Ocádiz and Juan Manuel Hurtado for their computer support, and María Luisa Tabche for her laboratory management.

## Author contributions

**Conceptualization:** Enrique Merino, Maricela Carrera-Reyna, Edna Cruz-Flores.

**Data curation:** Enrique Merino, Maricela Carrera-Reyna, Edna Cruz-Flores.

**Formal analysis:** Enrique Merino, Maricela Carrera-Reyna.

**Investigation:** Enrique Merino, Maricela Carrera-Reyna.

**Methodology:** Enrique Merino, Maricela Carrera-Reyna, Edna Cruz-Flores.

**Resources:** Enrique Merino.

**Software:** Maricela Carrera-Reyna, Edna Cruz-Flores.

**Supervision:** Enrique Merino.

**Validation:** Enrique Merino, Maricela Carrera-Reyna.

**Visualization:** Maricela Carrera-Reyna, Edna Cruz-Flores.

**Writing – original draft:** Enrique Merino, Maricela Carrera-Reyna, Edna Cruz-Flores.

**Writing – review & editing:** Enrique Merino, Maricela Carrera-Reyna, Edna Cruz-Flores.

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
