## [Decision Letter · Decision Letter 0]

10 Mar 2025

PONE-D-25-07489Analysis and comparison of the bacterial σ54 regulon: evidence of phylogenetic trends in gene regulationPLOS ONE

Dear Dr. Merino,

Thank you for submitting your manuscript to PLOS ONE. After careful consideration, we feel that it has merit but does not fully meet PLOS ONE’s publication criteria as it currently stands. Therefore, we invite you to submit a revised version of the manuscript that addresses the points raised during the review process.

We look forward to receiving your revised manuscript.

Kind regards,

Shengwei Sun, Ph.D.

Academic Editor

PLOS ONE

“Our work was supported by DGAPA-Universidad Nacional Autónoma de México (UNAM) PAPIIT (Grant IN222423); awarded to E.M. M.C. is a Ph.D. student enrolled in the Programa de Doctorado en Ciencias Bioquímicas-UNAM and recipient of a CONACYT fellowship (731966).”

Reviewers' comments:

Reviewer's Responses to Questions

**Comments to the Author**

1. Is the manuscript technically sound, and do the data support the conclusions?

Reviewer #1: Yes

Reviewer #2: Yes

2. Has the statistical analysis been performed appropriately and rigorously? 

Reviewer #1: No

Reviewer #2: Yes

3. Have the authors made all data underlying the findings in their manuscript fully available?

Reviewer #1: Yes

Reviewer #2: Yes

4. Is the manuscript presented in an intelligible fashion and written in standard English?

Reviewer #1: Yes

Reviewer #2: Yes

5. Review Comments to the Author

Reviewer #1: The authors present a study titled “Analysis and comparison of the bacterial σ54 regulon: evidence of phylogenetic trends in gene regulation.” This work provides a comprehensive analysis of the presence of Sigma54 binding sites across a diverse range of bacterial species. The results are original, as the authors effectively demonstrate the distribution of these regulons across phylogenetic classes using their own regulon identification method. This approach offers valuable insights into evolutionary trends in gene regulation.

1. Comprehensive and review style introduction to Sigma54.

2. l.200 The R script used to calculate COG overrepresentation is not provided. Please include it to ensure reproducibility.

3. l.207 Identification of EBPs is based on the presence of the PF00158 domain. However, it is unclear how specific this domain is. Could there be potential false positives? Provide an error analysis supporting the specificity of this domain

4. l.233 Please add the webpage link here: https://biocomputo.ibt.unam.mx/arpondb/. The web server is up and running, and the data is easily accessible.

5. l.240 Phylogenetic distribution of σ54: The phylogenetic group Actinomycetota appears to have only four organisms with a Sigma54. Could these 4 be false positives? How do these four organisms differ? Additionally, how confident are you that the AID, CBD, and DBD domains are not misidentified (are false positives)?

6. l.286 Identifying protein function based on a single domain is problematic. While the approach is generally valid, it is likely to result in a significant number of false positives and false negatives. How specific is the Pfam Sigma54_activat motif (PF00158)? The results indicate up to 100 EBPs per genome—how realistic is this estimate? See also point 3

7. l.321 The Matthew correlation coefficient (MCC) of 0.778 appears to be based on E. coli. Given the reported precision and specificity of 100%, this suggests an overfitted model for E. coli. How can such a model be reliably applied to genomes that are not closely related to E. coli? Error analysis is not provided

8. l.332 The values of the four parameters for the hypergeometric distribution depend heavily on the number of false positives and false negatives detected. If genomes from different taxonomic groups are included, the DNA and protein motifs will likely introduce more errors, which will impact the hypergeometric results. How confident are you that Sigma54 is being analyzed specifically, rather than general sequence diversity? Would a similar analysis with Sigma70 yield comparable trends? Was this tested?

9. l.1.4 – l.1.8 The list of pathways is extensive. While some have literature references to support their regulation by Sigma54, in many cases, it is unclear whether these pathways are directly regulated. Providing additional context or evidence would improve clarity.

10. l.477 The same issue applies to Cellular processes and signaling.

11. l.683 Similar concerns apply to Information storage and processing.

12. Several web servers and computational tools are already available for mining Sigma54 regulons. However, the authors do not provide a comparison to existing tools or demonstrate whether their method offers any improvements. A comparative analysis would strengthen the study.

13. A clear overview is missing that directly compares known regulon-associated genes/proteins with those identified in this study. Additionally, there is no clear indication of genes previously unknown to be influenced by Sigma54. Such a summary would enhance the impact of the findings.

14. l.834 "Through rigorous statistical analysis": The cross-organism application of PSSMs for motif identification in DNA is highly unreliable. The manuscript does not explain how errors were quantified or how confident the authors are in their results. Please provide details on this analysis.

15. l.840 "Our study revealed unreported cellular processes": Could these findings be due to false positives? Additionally, expected genes are not explicitly discussed—were they identified as expected, or were some missed? These results should be included to clarify the reliability of the method.

Reviewer #2: This is a very interesting survey of the sigma54 regulon across a very large set of genome sequences. It makes an important contribution to the field. The accompanying website is also a useful contribution.

However, I have some suggestions and comments that the authors may wish to consider.

The website at https://biocomputo.ibt.unam.mx/arpondb/ is a good contribution. However, previous experience suggests that websites often "disappear" after publication as the authors move on to new projects, funding for webhosting dries up, etc. I strongly urge the authors to deposit a snapshot of the webpage in Zotero or as supplementary data attached to the manuscript.

I am surprised that the authors do not cite this paper:

Barrios, H., Valderrama, B., & Morett, E. (1999). Compilation and analysis of sigma(54)-dependent promoter sequences. Nucleic acids research, 27(22), 4305–4313. https://doi.org/10.1093/nar/27.22.4305

This is a seminal paper that represents the first major attempt to systematically catalogue sigma54 promoter sequences and underpins much of the subsequent computer-based predictive studies of the sigma54 regulon.

The authors should also discuss this recent work and how it is similar/dissimilar the theirs:

Achterberg, T., & de Jong, A. (2025). ProPr54 web server: predicting σ54 promoters and regulon with a hybrid convolutional and recurrent deep neural network. NAR genomics and bioinformatics, 7(1), lqae188. https://doi.org/10.1093/nargab/lqae188

Lines 117-125. When selecting genome sequences for their analyses, did they consider the quality/completeness/reliability of the genome sequences? If yes, then what quality criteria did they apply. If not, then why not?

Lin 159. "propensity of some sigma factors for self-transcription". Is there any published evidence that sigma54 initiates transcription of itself? Given the uniqueness of this transcription factor, it is dangerous to extrapolate what we know about other sigma factors to sigma54. It would make more sense to start the seed matrix using a set of confirmed promoter sequences or a datasets such as that of Barrios et al. (1999).

Line 165. "cutoff value of 0.001". Value of what? Is this an E-value cut-off?

Line 247. Please be consistent in taxonomic names. Cereibacter sphaeroides is a synonym of Rhodobacter sphaeroides. So, use one name or the other. Don't use one synonym for one strain and another synonym for the other. Both names refer to precisely the same entity, so it makes no sense to use both. See https://lpsn.dsmz.de/species/rhodobacter-sphaeroides for full taxonomic information about this species.

Line 260. "Pfam motifs AID, DBD, and CBD". Are you sure that these are "motifs" in Pfam? Pfam has families, domains and clans. I don't think it has motifs. Also, please provide the Pfam accession numbers for these entities.

Line 273. "We hypothesize that this overlap in high GC content ...". To what extent has this been previously explored in the literature and to what extent does previous work support the hypothesis? For example, see:

Calistri, E., Livi, R., & Buiatti, M. (2011). Evolutionary trends of GC/AT distribution patterns in promoters. Molecular phylogenetics and evolution, 60(2), 228–235. https://doi.org/10.1016/j.ympev.2011.04.015

Line 277. "Tree of Life [with capital initial letters]". What is this? Can the authors provide a reference or URL? Do they mean https://www.evogeneao.com/en/explore/tree-of-life-explorer? Or https://itol.embl.de/?

Line 353. "Nitrogen metabolism has long been recognized as one of the key processes that is regulated by σ54, initially leading to the association of this sigma factor with the name σN. However, our study's findings challenge the notion that the regulation of nitrogen metabolism, which is indicated by the red circles in Fig 5, is a universal trend across bacteria.". Here, the authors are setting up a "straw man". I am not aware of any recent reviews that posit that sigma54's role in nitrogen regulation is universal. Can the authors cite any specific papers with which they disagree on this point?

Line 403. "Interestingly, as previously documented, RNAP-σ54 complex also regulates the transcription of PTS component genes in β-Proteobacteria. Also gamma-proteobacteria.

See:

Studholme D. J. (2002). Enhancer-dependent transcription in Salmonella enterica Typhimurium: new members of the sigmaN regulon inferred from protein sequence homology and predicted promoter sites. Journal of molecular microbiology and biotechnology, 4(4), 367–374.

Line 507. When considering the sigma54 regulon in planctomycetes, the authors may wish to read this paper:

Studholme, D. J., & Dixon, R. (2004). In silico analysis of the sigma54-dependent enhancer-binding proteins in Pirellula species strain 1. FEMS microbiology letters, 230(2), 215–225. https://doi.org/10.1016/S0378-1097(03)00897-8.

I note that the authors' predictions at https://biocomputo.ibt.unam.mx/arpondb/tables/rba_Rhodopirellula_baltica.html are not entirely consistent with the results in that paper. The authors may or may not wish to consider that. Furthermore, I was unable to locate in the authors' web database their predictions for "Aquifex aeolicus" to compare those against the published work the sigma54 regulon in that organism.

Throughout the manuscript, please use a superscript for 54 when writing sigma54. And do the same for other sigma factors; e.g. sigma70 should have superscripted 70.

Line 23 and elsewhere. Rather than saying "bacterial organisms", it is more straightforward to just say "bacteria".

Line 25. Instead of "corresponds to" simply write "is".

Line 33. Delete "lacking sequence homology". First, it is unclear how sequence homology is different from simply "homology". Second, the lack of homology is discussed in subsequent sentences, so does not need to be stated here.

Line 38. The word "instances" does not make sense here. Please consider using a different word.

Line 44. "ubiquitously present". No. As the authors show later in this manuscript, this sigma factor is NOT ubiquitous. There are several bacterial taxa where this sigma factor is absent.

Line 57. "class Clostridiales". This is an order, not a class.

Line 79. "matrices.". These are not matrices. They are profile-HMMs.

Line 78. Do not use italics for Pfam.

Line 112. "1,249 organisms". Here, and many other times in the manuscript the authors refer to numbers of organisms. What does this mean? Does it mean numbers of species? Strains? How do the authors count organisms? Are two strains counted as two different organisms?

Line 289. The use of italics has gone wrong on this line.

Line 481. Use lowercase 'c' for cellular.

6. PLOS authors have the option to publish the peer review history of their article (what does this mean? ). If published, this will include your full peer review and any attached files.

**Do you want your identity to be public for this peer review?** For information about this choice, including consent withdrawal, please see our Privacy Policy .

Reviewer #1: No

Reviewer #2: **Yes: ** David Studholme

---

## [Author Response · Author response to Decision Letter 1]

6 Jun 2025

Dear Editor,

We sincerely thank you and the reviewers for your thoughtful and constructive comments. Your feedback was instrumental in improving the clarity, depth, and overall quality of our manuscript. We have carefully considered each suggestion and made the corresponding revisions, which we believe have significantly strengthened the work. We are grateful for the time and effort you devoted to the review process and for helping us enhance the impact of our study.

Response: We have verified that the submission complies with PLOS ONE’s style requirements, including those related to file naming.2. Thank you for stating the following financial disclosure:

“Our work was supported by DGAPA-Universidad Nacional Autónoma de México (UNAM) PAPIIT (Grant IN222423); awarded to E.M. M.C. is a Ph.D. student enrolled in the Programa de Doctorado en Ciencias Bioquímicas-UNAM and recipient of a CONACYT fellowship (731966).”

Response: In the revised version of our cover letter, we have added the following statement: “The funders had no role in study design, data collection and analysis, decision to publish, or preparation of the manuscript.”

Response: The phrase “data not shown” appeared only once in the previous version of our manuscript. As recommended, we have removed it, since the referenced data were not essential to the main findings of the article (see line 336 of our track-changes document).

Reviewers' comments:

Reviewer's Responses to Questions

Comments to the Author

1. Is the manuscript technically sound, and do the data support the conclusions?

Reviewer #1: Yes

Reviewer #2: Yes

2. Has the statistical analysis been performed appropriately and rigorously?

Reviewer #1: No

Reviewer #2: Yes

3. Have the authors made all data underlying the findings in their manuscript fully available?

Reviewer #1: Yes

Reviewer #2: Yes

4. Is the manuscript presented in an intelligible fashion and written in standard English?

Reviewer #1: Yes

Reviewer #2: Yes

5. Review Comments to the Author

Reviewer #1: The authors present a study titled “Analysis and comparison of the bacterial σ54 regulon: evidence of phylogenetic trends in gene regulation.” This work provides a comprehensive analysis of the presence of Sigma54 binding sites across a diverse range of bacterial species. The results are original, as the authors effectively demonstrate the distribution of these regulons across phylogenetic classes using their own regulon identification method. This approach offers valuable insights into evolutionary trends in gene regulation.

1. Comprehensive and review style introduction to Sigma54.

Response: We have expanded the Introduction section of our manuscript to provide a more comprehensive and review-style overview of σ⁵⁴. Additionally, we have carefully revised its structure and writing style to enhance clarity and coherence.

2. l.200 The R script used to calculate COG overrepresentation is not provided. Please include it to ensure reproducibility.

Response: To evaluate the statistical significance of the observed enrichment of genes regulated by σ⁵⁴ in specific COG categories across different phylogenetic clades, we used a custom PERL script that utilizes the phyper and dhyper functions from the R package. This script is publicly available via a GitHub repository, as referenced in the Methods section (line 221 of our track-changes document).

3. l.207 Identification of EBPs is based on the presence of the PF00158 domain. However, it is unclear how specific this domain is. Could there be potential false positives? Provide an error analysis supporting the specificity of this domain

In response to the referee’s concern regarding the specificity of our EBP identification results, we conducted a more detailed inspection of the sequences initially identified as true positives in our hmmsearch analysis using the Pfam domain PF00158 (Sigma54_activat). We found that, despite strong sequence conservation and statistically significant E-values, some sequences lacked the “GAFTGA” motif, which is essential for interaction with the σ⁵⁴-RNA polymerase complex (39, 40).

Based on this observation, we adopted a more stringent approach to eliminate false positives from our EBP dataset. This revised pipeline integrates both, domain-based identification (via PF00158) and domain architecture analysis, as described in the Identification of σ54-specific EBPs of the Methods section. The steps are as follows:

1. Initial EBP Identification: we searched for the Pfam domain PF00158 (Sigma54_activat) within the Pfam database. This domain is known to be conserved in EBP proteins, which are involved in ATP-dependent interactions with σ54. In our search, we employed an E-value threshold of 1e-51 to ensure the inclusion of EBP candidates with high confidence.

2. Sequence Redundancy Reduction: The sequences containing PF00158 identified in the previous step were clustered using CD-HIT program (62).

3. Motif Discovery: The resulting non-redundant sequences were analyzed using MEME to identify conserved motifs.

4. Motif Scanning Across Proteomes: We then used MAST to scan for the presence of the 10 motifs discovered by MEME in the full set of proteomes.

5. Filtering of sequences for the presence of the GAFTGA motif. Considering that the “GAFTGA” motif have been found in functionally EBP proteins, we select all sequences containing this essential motif.

6. Final Filtering of High-Confidence σ⁵⁴-specific EBPs: Finally, we parsed the MAST output to retain only those sequences that contained all 10 motifs identified in the step #3 arranged in the following conserved order:

(2) – (10) – (6) – (3) – (4) – (9) – (7) – (5) – (8) – (1)

Sequences that meet all the previous filters were considered as bona-fide σ⁵⁴-specific EBP proteins.

As a negative control for our protocol to identify σ⁵⁴-specific EBP proteins, we analyzed the proteomes of 380 Archaea and 36 Eukaryota (Fungi), comprising a total of 1,169,896 proteins. Our results demonstrate 100% specificity, as no σ⁵⁴-specific EBPs were detected within this negative control set.

Based on our new results, we have updated the Identification of EBPs (lines 229–255) and the Phylogenetic Distribution of EBPs (lines 291 to 307) sections in the revised version of our article. The list of organisms used as a negative control is provided in Supplementary Table S4.

4. l.233 Please add the webpage link here: https://biocomputo.ibt.unam.mx/arpondb/. The web server is up and running, and the data is easily accessible.

We have confirmed that the web server is functioning as expected. Additionally, we have included the link to our webpage at the end of the Abstract section (lines 19 and 20 of our track-changes document).

5. l.240 Phylogenetic distribution of σ54: The phylogenetic group Actinomycetota appears to have only four organisms with a Sigma54. Could these 4 be false positives? How do these four organisms differ? Additionally, how confident are you that the AID, CBD, and DBD domains are not misidentified (are false positives)?

In response to the referee’s concern regarding the authenticity of the four σ⁵⁴-encoding organisms, we performed a multiple sequence alignment of their σ⁵⁴ proteins alongside those from four well-characterized reference species. To quantitatively assess sequence conservation, we calculated pairwise similarities between the AID, CBD, and DBD domains of the Actinomycetota sequences and the reference σ⁵⁴ proteins. The resulting similarity values are compiled into square matrices and presented in a new supplementary table (S6 Table). In addition, comparable matrices based on Pfam domain bitscores are provided in S7 Table.

A careful examination of the data revealed that, among the original set of four Actinomycetota organisms with putative RpoN proteins, only Arthrobacter citreus NEB 577 encodes an RpoN protein with significant Pfam bit scores across all three conserved domains (AID, CBD, and DBD). Further analyses, including the construction of a 16S rRNA-based phylogenetic tree and a detailed review of NCBI taxonomic annotations, suggested a likely misclassification: Arthrobacter citreus NEB 577, originally assigned to Actinomycetota, appears to be more closely related with the class Bacilli.

Taken together, our findings support the conclusion that Actinomycetota, as currently classified, lacks a functional σ⁵⁴ sigma factor. Accordingly, we have revised the Phylogenetic distribution of σ54 of the Results and Discussion section (lines 316 to 327 of our track-changes document).

6. l.286 Identifying protein function based on a single domain is problematic. While the approach is generally valid, it is likely to result in a significant number of false positives and false negatives. How specific is the Pfam Sigma54_activat motif (PF00158)? The results indicate up to 100 EBPs per genome—how realistic is this estimate? See also point 3

As explained in our response to comment 3, which raised a similar concern about the limited confidence in identifying σ⁵⁴ proteins based solely on a single Pfam domain, we have refined our search strategy. The revised analysis now employs a more rigorous approach that incorporates ten distinct motifs identified using the MEME program, filters sequences for the presence of the GAFTGA motif, and performs a comprehensive search across the proteomes of our non-redundant organism set. This updated procedure is detailed in the Identification of σ⁵⁴-specific EBPs (lines 229–255) of the Methods section, and Phylogenetic Distribution of σ⁵⁴-specific EBPs (lines 316–327) of the Results and Discussion section, of the revised manuscript.

7. l.321 The Matthew correlation coefficient (MCC) of 0.778 appears to be based on E. coli. Given the reported precision and specificity of 100%, this suggests an overfitted model for E. coli. How can such a model be reliably applied to genomes that are not closely related to E. coli? Error analysis is not provided

In response to the referee’s comment, we adopted a revised strategy to minimize potential bias toward E. coli or other model organisms in the construction of the initial seed matrix used at the onset of our iterative promoter search procedure. Specifically, we curated a diverse dataset of RpoN-dependent promoter sequences from the most frequently cited scientific publications and major publicly available regulatory databases. This dataset comprises 720 experimentally validated RpoN promoter sequences from 56 different organisms obtained from major regulatory databases, including RegPrecise [56], DBTBS [57], EcoCyc [58], and from the compilation of σ⁵⁴-dependent promoter sequences reported in Barrios et al. (1999) [59].

The initial set of RpoN promoter sequences was used to construct a seed position-specific scoring matrix (PSSM) using MEME. We then implemented an iterative motif discovery and refinement process, combining MEME and MAST. The seed PSSM was used in the initial MAST search, and in subsequent rounds, it was updated with matrices derived from taxon-specific sequences identified at each step. As noted previously, this cyclic procedure was applied independently to each of the 33 phylogenetic classes to enhance precision. We believe that this design, grounded in promoter data from phylogenetically diverse organisms, substantially reduces the risk of overfitting.

To compare the performance of our original protocol with the updated version, we used the same set of experimentally verified σ⁵⁴ promoters reported in EcoCyc. Both approaches yielded similar results for the Matthews correlation coefficient (0.778 vs 0.778), specificity (100% vs 100%), and sensitivity (60% vs 60%). As in our previous analysis, we attribute the high specificity of the predictions to the stringent significance thresholds applied, regardless of how the seed matrix was constructed. We believe that this high specificity ensured the retention of only high-confidence σ⁵⁴ promoter sequences as true positives.

In the Methods section (lines 173–190 of our track-changes document), we updated the description of the procedure used to generate the seed PSSM and made minor modifications to Fig 2 to reflect the improvements introduced at this initial step. Additionally, we now include the curated list of RpoN-dependent promoter sequences used in our analysis as a new supplementary table (Table S2), as well as the seed position-specific scoring matrices (PSSMs) used for the initial MAST searches in each of the taxonomic groups analyzed (Table S3). Furthermore, we revised lines 377–378 of the track-changes document to report the performance of our computational protocol using a new seed matrix generated from experimentally validated σ⁵⁴ promoter sequences in Escherichia coli and Bacillus subtilis, representing the Pseudomonadota and Bacillota phyla, respectively.

8. l.332 The values of the four parameters for the hypergeometric distribution depend heavily on the number of false positives and false negatives detected. If genomes from different taxonomic groups are included, the DNA and protein motifs will likely introduce more errors, which will impact the hy

---

## [Decision Letter · Decision Letter 1]

23 Jun 2025

Analysis and comparison of the bacterial σ54 regulon: evidence of phylogenetic trends in gene regulation

PONE-D-25-07489R1

Dear Dr. Merino,

We’re pleased to inform you that your manuscript has been judged scientifically suitable for publication and will be formally accepted for publication once it meets all outstanding technical requirements.

Kind regards,

Shengwei Sun, Ph.D.

Academic Editor

PLOS ONE

Additional Editor Comments (optional):

Reviewers' comments:

Reviewer's Responses to Questions

**Comments to the Author**

1. If the authors have adequately addressed your comments raised in a previous round of review and you feel that this manuscript is now acceptable for publication, you may indicate that here to bypass the “Comments to the Author” section, enter your conflict of interest statement in the “Confidential to Editor” section, and submit your "Accept" recommendation.

Reviewer #1: All comments have been addressed

Reviewer #2: All comments have been addressed

2. Is the manuscript technically sound, and do the data support the conclusions?

Reviewer #1: Yes

Reviewer #2: Yes

3. Has the statistical analysis been performed appropriately and rigorously? 

Reviewer #1: Yes

Reviewer #2: I Don't Know

4. Have the authors made all data underlying the findings in their manuscript fully available?

Reviewer #1: Yes

Reviewer #2: Yes

5. Is the manuscript presented in an intelligible fashion and written in standard English?

Reviewer #1: Yes

Reviewer #2: Yes

6. Review Comments to the Author

Reviewer #1: The authors have significantly improved the quality of the manuscript. In particular, the refinement of the search strategy and the inclusion of new supplementary data on phylogenetic distribution have strengthened the study and made the manuscript suitable for publication.

Reviewer #2: (No Response)

7. PLOS authors have the option to publish the peer review history of their article (what does this mean? ). If published, this will include your full peer review and any attached files.

**Do you want your identity to be public for this peer review?** For information about this choice, including consent withdrawal, please see our Privacy Policy .

Reviewer #1: No

Reviewer #2: **Yes: ** David Studholme

---

## [Editor Report · Acceptance letter]

PONE-D-25-07489R1

PLOS ONE

Dear Dr. Merino,

I'm pleased to inform you that your manuscript has been deemed suitable for publication in PLOS ONE. Congratulations! Your manuscript is now being handed over to our production team.

Kind regards,

on behalf of

Dr. Shengwei Sun

Academic Editor

PLOS ONE